# Intranasal or airborne transmission-mediated delivery of an attenuated SARS-CoV-2 protects Syrian hamsters against new variants

Charles B. Stauft[1,4], Prabhuanand Selvaraj [1,4], Felice D'Agnillo [2,4], Clement A. Meseda[1,4], Shufeng Liu [1], Cyntia L. Pedro[1], Kotou Sangare[1], Christopher Z. Lien [1], Jerry P. Weir[1], Matthew F. Starost[3] & Tony T. Wang [1] ✉

Detection of secretory antibodies in the airway is highly desirable when evaluating mucosal protection by vaccines against a respiratory virus, such as the severe acute respiratory syndrome coronavirus 2 (SARS-CoV-2). We show that intranasal delivery of an attenuated SARS-CoV-2 (Nsp1-K164A/H165A) induces both mucosal and systemic IgA and IgG in male Syrian hamsters. Interestingly, either direct intranasal immunization or airborne transmission-mediated delivery of Nsp1-K164A/H165A in Syrian hamsters offers protection against heterologous challenge with variants of concern (VOCs) including Delta, Omicron BA.1, BA.2.12.1 and BA.5. Vaccinated animals show significant reduction in both tissue viral loads and lung inflammation. Similarly attenuated viruses bearing BA.1 and BA.5 spike boost variant-specific neutralizing antibodies in male mice that were first vaccinated with modified vaccinia virus Ankara vectors (MVA) expressing full-length WA1/2020 Spike protein. Together, these results demonstrate that our attenuated virus may be a promising nasal vaccine candidate for boosting mucosal immunity against future SARS-CoV-2 VOCs.

The early success of the two mRNA vaccines against SARS-CoV-2 boasted up to 95% efficacy against the most severe disease outcomes[1,2]. Nevertheless, the emergence of new virus variants has resulted in reduced vaccine effectiveness against symptomatic COVID-19. Since the emergence of the Omicron variant of concern (VOC) in late 2021, SARS-CoV-2 Omicron sub-lineages (e.g., BA.1, BA.2, BA.3, BA.4, BA.5, BQ.1, XBB, etc.) have gradually replaced previously circulating variants worldwide[3]. Owing to increased immune evasion, infection with Omicron has led to large numbers of hospitalizations and deaths[4–6]. Vaccine effectiveness against Omicron after two BNT162b2 doses was modest at 2–4 weeks and fell to nearly zero after 6 months or more[7–9].

A third and fourth mRNA vaccine booster did improve efficacy against Omicron to 60–70%[10–12], but the durability of protection was less than impressive, with a mean 30-day rate of decay in neutralizing antibody titers of nearly 20% against BA.4/5[13].

Several studies have suggested that COVID-19 vaccine performance might be improved if mucosal immunity can be enhanced. For example, a recent study demonstrated that combining systemic mRNA vaccination with mucosal adenovirus-Spike immunization induced strong neutralizing antibody (nAb) responses against both the ancestral virus and the Omicron BA.1.1 variant in mice. By contrast, systemic mRNA vaccination alone induced weak respiratory mucosal

[1]Division of Viral Products, Center for Biologics Evaluation and Research, Food and Drug Administration, Silver Spring, MD, USA. [2]Laboratory of Biochemistry and Vascular Biology, Center for Biologics Evaluation and Research, Food and Drug Administration, Silver Spring, MD, USA. [3]Division of Veterinary Resources, Diagnostic and Research Services Branch, National Institutes of Health, Rockville Pike, MD, USA. [4]These authors contributed equally: Charles B. Stauft, Prabhuanand Selvaraj, Felice D'Agnillo, Clement A. Meseda. ✉e-mail: Tony.Wang@fda.hhs.gov

neutralizing antibody responses[14]. Earlier in another proof-of-concept study, an intranasal vaccination with nonadjuvanted spike subunit protein following intramuscular mRNA vaccinations in mice elicited protective mucosal immunity via memory T/B cells and IgA that significantly lowered viral load in the upper and lower airways and prevented disease and death from a lethal SARS-CoV-2 challenge[15]. These findings highlight the importance of understanding the mucosal immunogenicity and efficacy of next-generation nasal vaccines[16].

Of the many vaccines in development against SARS-CoV-2, live attenuated virus (LAV) vaccines are a substantial minority despite the potential for nasal administration and the advantage of presenting all viral antigens to the host immune system[17–19]. To facilitate the evaluation of LAVs, we recently developed a genetic approach to attenuate SARS-CoV-2. Our strategy consists of three attenuating modifications to the viral genome: the removal of the furin cleavage site, the deletion

of ORFs 6–8[20], and introduction of a pair of mutations to the Nsp1 gene (Fig. 1a)[21]. The resulted WA1-ΔPRRA-ΔORF6-8-Nsp1$^{K164A/H165A}$ (abbreviated as Nsp1-K164A/H165A in this paper) is attenuated both in vitro and in vivo compared to wild-type virus and was immunogenic and protective against the ancestral SARS-CoV-2 challenge[22]. In the current study, we assessed the mucosal immunogenicity, efficacy in protecting disease caused by recent variants of concern, possibility as a variant-specific booster, as well as the transmission of this attenuated SARS-CoV-2 vaccine candidate.

## Results

### Mucosal and systemic immunogenicity of Nsp1-K164A/H165A

First, we assessed anti-SARS-CoV-2 spike Immunoglobulin G (IgG) and IgA in serum samples and nasal washes from Syrian hamsters after intranasal inoculation of 100 PFU Nsp1-K164A/H165A or the wild-type

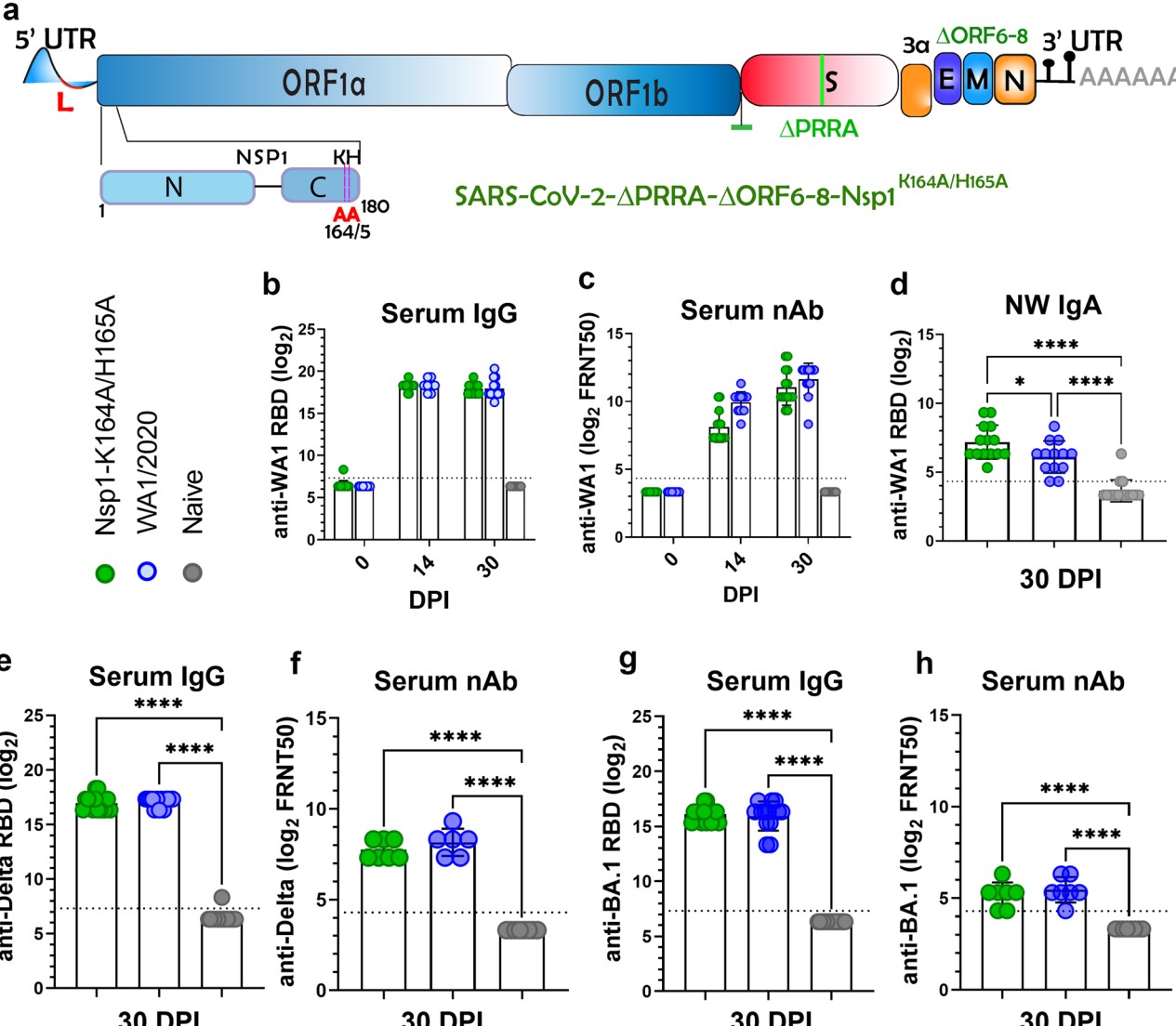

**Fig. 1 | Intranasal immunization with 100 PFU Nsp1-K164A/H165A induces IgG and IgA. a** Genome organization of the attenuated SARS-CoV-2-ΔPRRA-ΔORF6-8-Nsp1$^{K164A/H165A}$ (abbreviated as Nsp1-K164A/H165A). The polybasic insert "PRRA" together with ORF6-8(green) were removed from the WA1/2020 genome. Locations of K164A/H165A within Nsp1 are indicated at the bottom left of the panel. **b–h** Hamster sera or nasal wash samples were collected at 14- and 30-days post-intranasal inoculation of 100 PFU Nsp1-K164A/H165A or WA1/2020 and then tested for binding to WA1-2020 receptor binding domain (RBD) by ELISA (**b**) and for neutralization against WA1/2020 (**c**). Samples from naïve hamsters were included as negative controls. **d** Secretory IgA levels in nasal wash samples (NW IgA)

collected 30 DPI were measured by ELISA. *$p = 0.0302$. **e–h** Hamster sera of 30 DPI were measured for anti-Delta RBD IgG (**e**) and neutralization (**f**), for anti-Omicron BA.1 RBD IgG (**g**), and neutralization (**h**). Biologically independent samples for WA1/2020 ($n = 13$), Nsp1-K164A/H165A ($n = 14$), and naïve hamsters ($n = 16$) were used in a single independent experiment. Bar graphs indicate mean titers with standard deviations shown as error bars. Each solid circle indicates individual hamsters from a single experiment. Statistical differences were calculated using ordinary one-way analysis of variance (ANOVA) in GraphPad Prism 9.4.0 with Tukey's multiple comparisons tests. For statistical significance, *$p < 0.05$ and ****$p < 0.0001$. DPI days post-infection.

WA1/2020 virus. Sera collected at 14- and 30-days post-infection (DPI) from both groups contained high titers of IgG antibodies specific for the WA1/2020 receptor binding domain (RBD) (Fig. 1b). Serum nAb titers increased from 0 to 14 DPI ($p < 0.0001$, mixed-effects analysis) and then from 14 to 30 DPI ($p < 0.0001$) for both groups of animals. Geometric mean titers (GMT) of serum nAbs against WA1/2020, indicated by 50% focus forming reduction neutralization titers ($FRNT_{50}$), were 2100 (interquartile range, IQR, 3840) and 3169 (IQR 2560) at 30 DPI for Nsp1-K164A/H165A and WA1/2020 groups, respectively (Fig. 1c). At 30 DPI, IgA titers detected in nasal washes were significantly higher ($p < 0.0001$) in Nsp1-K164A/H165A (GMT 145, IQR 240) and WA1/2020 (GMT 68, IQR 80) groups compared to naive controls (Fig. 1d). In this experiment, there was 2-fold higher IgA titers ($p = 0.0336$) in the Nsp1-K164A/H165A group than in WA1/2020 infected animals. When the same sera were measured for anti-Delta variant RBD IgG, a 2-fold reduction was observed in both the WA1/2020 (GMT 139621 versus 250997) and Nsp1-K164A/H165A (GMT 127911 versus 243465) groups in comparison to WA1/2020 RBD-specific IgG titers (Fig. 1e in comparison to Fig. 1b). Concomitantly, serum nAb titers against Delta variant dropped by near 10-fold compared to those against the ancestral WA1/2020 for both Nsp1-K164A/H165A (GMT 215, IQR 160) and WA1/2020 (GMT 285, IQR 240) groups (Fig. 1f in

comparison to Fig. 1c). Similarly, serum Omicron BA.1 RBD-specific IgG titers, in comparison to WA1/2020 RBD-specific IgG titers, decreased by nearly 4-fold in Nsp1-K164A/H165A-vaccinated animals (GMT 243465 versus 70613) and in WA1/2020-infected (GMT 250997 versus 62749) hamsters at 30 DPI (Fig. 1g compared to Fig. 1b). NAb against the BA.1 variant reached titers just above the limit of detection from both the Nsp1-K164A/H165A-vaccinated group (GMT 36, IQR 20 i.e., a > 50-fold reduction compared to nAb titers against the ancestral WA1/2020) and the WA1/2020-infected group (GMT 44.2, IQR 40, i.e, a > 70-fold reduction compared to nAb titers against the ancestral WA1/2020) (Fig. 1h in comparison to Fig. 1c).

To ensure that the loss of serum neutralization antibody to Omicron BA.1 was not due to the low vaccine dose (100 PFU), we inoculated Syrian hamsters with $10^4$ PFU Nsp1-K164A/H165A and characterized antibody responses to various Omicron subvariants. Because secretory IgA (SIgA) is critical to antiviral immunity in the lung[23], we first assessed anti-SARS-CoV-2 spike IgG and IgA in serum and then in bronchoalveolar lavage fluid (BALF). At 14 and 28 DPI, serum anti-RBD IgG titers against WA1/2020 (GMT of 137772 and 163840, respectively) were 20-fold higher ($p < 0.0001$) than those against BA.1 variant (GMT of 3044 and 12177, respectively) (Fig. 2a). BALF collected on 14 and 28 DPI contained detectable levels of IgG specific for ancestral RBD (GMT

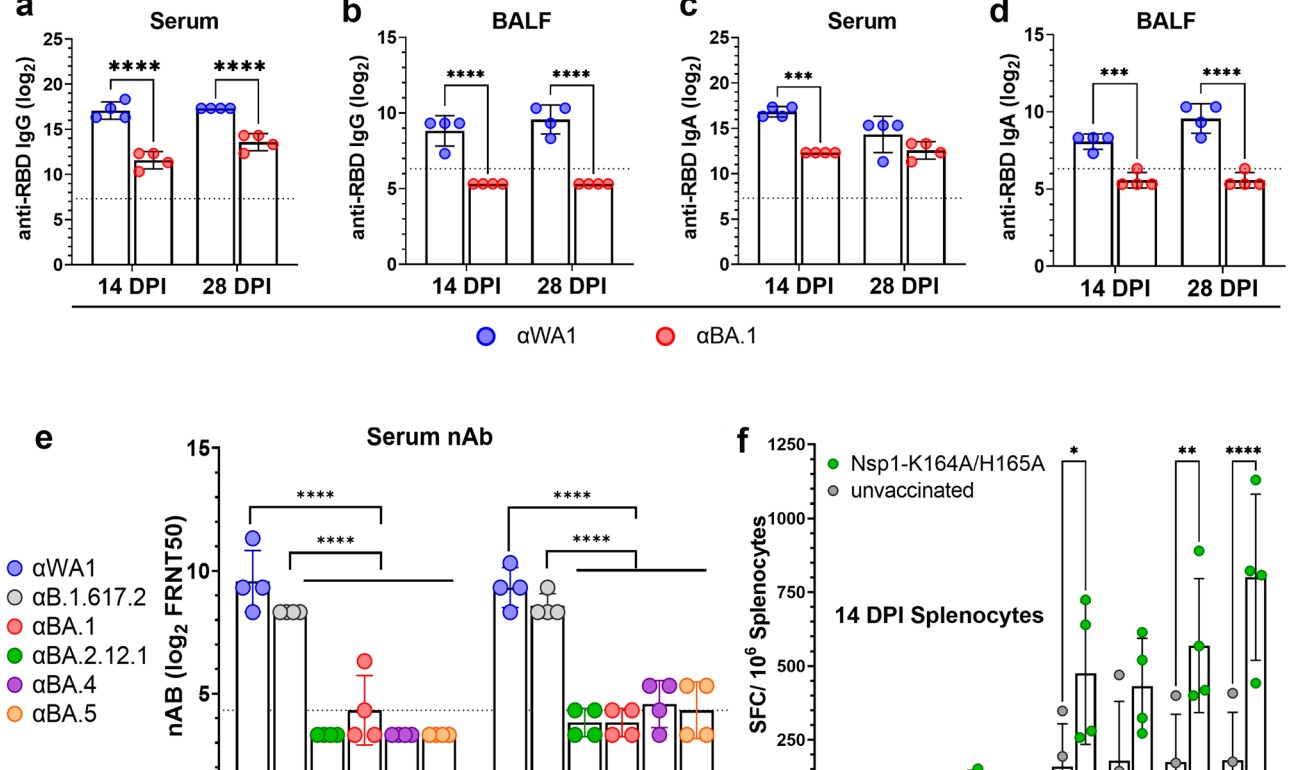

**Fig. 2 | Intranasal immunization with Nsp1-K164A/H165A induces mucosal and systemic humoral immunity and cellular immunity.** Male (4-month-old, $n = 8$) Syrian hamsters were intranasally vaccinated with $10^4$ PFU Nsp1-K164A/H165A. Animals were bled at 14 and 28 DPI to collect sera and $n = 4$ animals were euthanized at each time point to collect broncho-alveolar lavage fluid (BALF). Anti-RBD IgG titers against WA1/2020 (blue) or BA.1 (red) in serum (**a**) and BALF (**b**) were measured by ELISA. Anti-RBD IgA (***$p = 0.0002$) was likewise quantified in serum (**c**) and BALF (***$p = 0.0003$) (**d**). **e** Serum nAb titers at 28 DPI were measured against WA1/2020 (WA1), Delta (B.1.617.2), as well as Omicron subvariants (BA.1, BA.2.12.1, BA.4, BA.5) using a 50% focus reduction neutralization (FRNT50) assay.

**f** Splenocytes were isolated from naive and vaccinated hamsters at 14 DPI and pulsed with 10 μM WA1/2020 Spike (S) and Nucleocapsid (N) antigen pools for 48 h. IFNγ-secreting splenocytes were enumerated by ELISPOT. Bar graphs represent mean values with standard deviation for samples collected at two time points from the same animals in a single experiment with dots representing individual animals. *$p = 0.0246$, **$p = 0.0026$. ELISPOT data were compared using two-way ANOVA with Sidak's multiple comparison test. Unless otherwise indicated, statistical differences were calculated using ordinary one-way analysis of variance (ANOVA) in GraphPad Prism 9.4.0 with Tukey's multiple comparisons tests. For statistical significance, *$p < 0.05$, **$p < 0.01$, and ****$p < 0.0001$. DPI days post-infection.

of 452 and 761, respectively), however, IgG titers specific for BA.1 RBD were below the limit of detection (Fig. 2b). Serum WA1/2020 RBD-specific IgA titers decreased by 5.65-fold from 14 (GMT 115852, IQR 81920) to 28 DPI (GMT 20480, IQR 28800) ($p = 0.0233$, Sidak's multiple comparisons test), whereas serum anti-BA.1 RBD-specific IgA titers stayed at comparable levels at 14 and 28 DPI (Fig. 2c). In BALF, BA.1 RBD-specific IgA was undetectable at 14 and 28 DPI, but WA1/2020 RBD-specific IgA was detected at both 14 DPI ($p = 0.0003$) and 28 DPI ($p < 0.0001$) (Fig. 2d). GMT nAb titers against WA1/2020 were 761 (IQR 1680, 14 DPI) and 640 (IQR 720, 28 DPI) with greater than 32-fold reductions when measured against Omicron subvariants BA.1, BA.2.12.1, BA.4, and BA.5 (Fig. 2e). Lastly, we assessed the cellular immunity elicited by Nsp1-K164A/H165A vaccination. Significant induction of IFNγ-secreting cells was observed at 14 DPI in splenocytes harvested from vaccinated group pulsed with nucleocapsid antigen pools N1 ($p = 0.0096$), N3 ($p = 0.0096$), and N4 ($p < 0.0001$) by ELISpot assays (Fig. 2f). Taken together, a single dose of Nsp1-K164A/H165A, when administered intranasally, induced IgA/IgG against SARS-CoV-2 spike protein in both respiratory tract and in circulation. However,

these anti-Spike antibodies are variant-specific and subject to evasion by Omicron variants.

## Efficacy of Nsp1-K164A/H165A against challenge with Delta and Omicron variants

To assess whether intranasal administration of Nsp1-K164A/H165A offers protection against VOCs, male, 5-month-old Syrian hamsters were inoculated with 100 PFU of Nsp1-K164A/H165A ($n = 14$) or WA1/2020 ($n = 14$) by the intranasal route (derived from the study depicted in Fig. 1). At 35 days-post-infection (DPI), the vaccinated and convalescent hamsters ($n = 6$–7 per group) along with unvaccinated naïve hamsters ($n = 8$) were challenged with $10^4$ PFU of a Delta isolate (hCoV-19/USA/MD-HP05647/2021) or a BA.1 Omicron isolate (hCoV-19/USA/HI-CDC-4359259-001/2021). On 4 and 7 days-post-challenge (DPC), animals were euthanized to collect tissues for analyses of viral replication and pathology (Fig. 3a). Nasal wash samples were also collected from each hamster following challenge. We noted a significant reduction of infectious virus ($p < 0.0001$) in nasal wash samples at 2, 3, 4, and 5 DPC by Delta in animals inoculated with Nsp1-K164A/H165A or

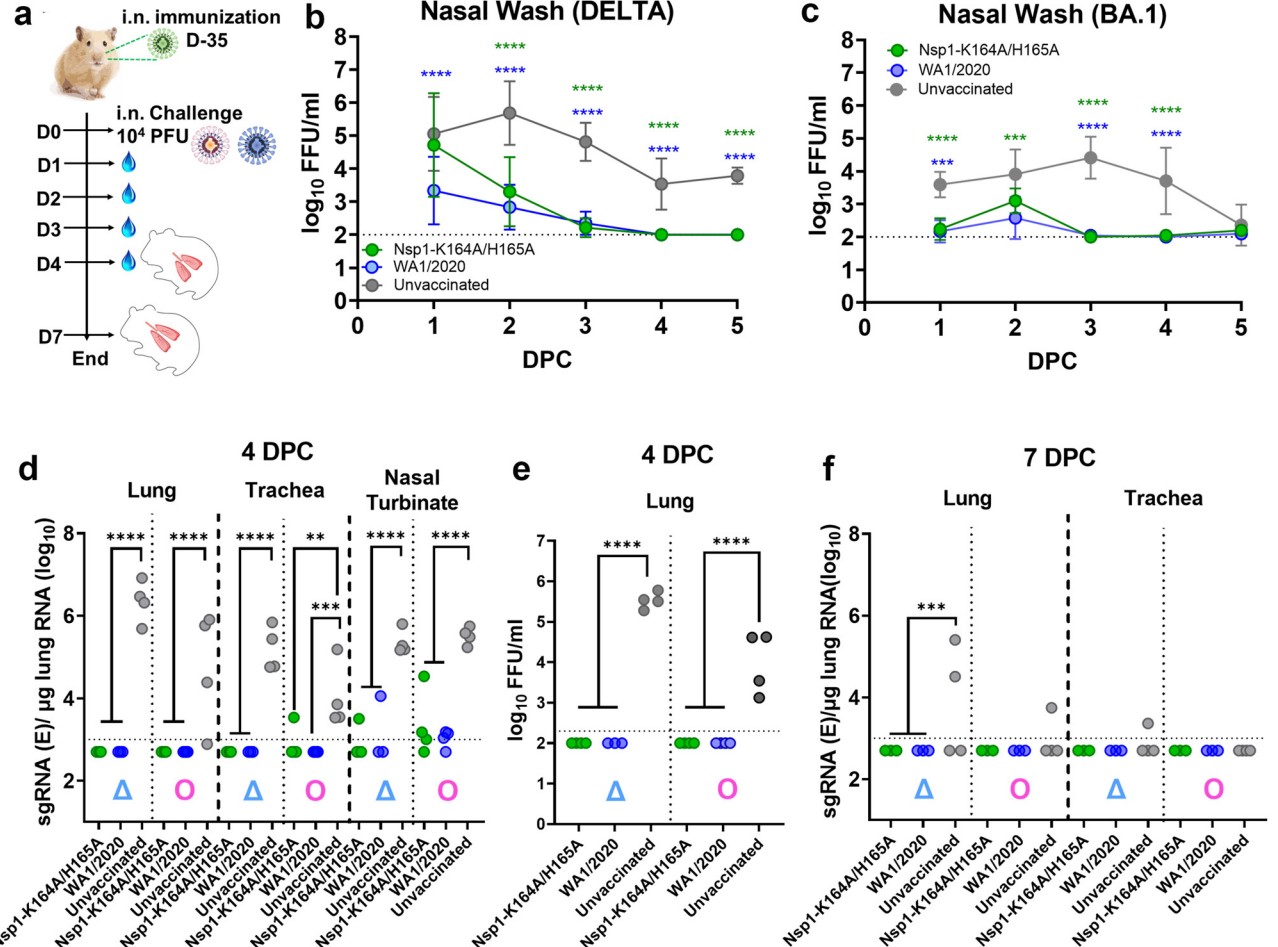

**Fig. 3 | Intranasal immunization of Syrian hamsters with 100 PFU Nsp1-K164A/H165A significantly reduces viral loads in both upper and lower respiratory tract following challenge with Delta and Omicron variants. a** Syrian hamsters were vaccinated with 100 PFU Nsp1-K164A/H165A or infected with 100 PFU WA1/2020 35 days prior to challenge with $10^4$ PFU Delta ($n = 6$) or BA.1 Omicron ($n = 7$) isolates on day 0. **b**–**c** From 1–5 DPC, infectious virus from nasal wash samples was quantified by focus-forming assays in vaccinated, convalescent, or unvaccinated hamsters ($n = 8$ for Delta, **b**; $n = 7$ for Omicron BA.1, **c**). Graphs for **b** and **c** indicate mean values from a single experiment with standard deviations shown as error bars. ***$p = 0.001$. **d** Viral sgRNA levels in lung, trachea, and nasal turbinate samples from 4 DPC ($n = 4$ each, except WA1/2020-Delta at $n = 3$) were measured by qRT-

PCR. **$p = 0.0041$, ***$p = 0.0008$. **e** Infectious virus titers of lung homogenates at 4 DPC were determined by focus-forming assays. **f** Viral sgRNA levels in lungs and trachea at 7 DPC with Delta and BA.1 Omicron were measured by qRT-PCR. Infectious and qRT-PCR based titrations of Nsp1-K164A/H165A ($n = 3$), WA1/2020 ($n = n = 3$) vaccinated, or unvaccinated ($n = 4$) biological replicates in one independent challenge experiment each for Delta and Omicron BA.1. ***$p = 0.0052$. Dot plots represent samples collected from individual animals in a single experiment. Statistical differences were calculated using ordinary one-way analysis of variance (ANOVA) in GraphPad Prism 9.4.0 with Tukey's multiple comparisons tests. For statistical significance, ****$p < 0.0001$. Δ delta variant challenge, O omicron BA.1 challenge, DPC days post-challenge.

WA1/2020 compared to unvaccinated controls (Fig. 3b). Similarly, Nsp1-K164A/H165A vaccinated and WA1/2020 convalescent hamsters also exhibited reduced nasal viral load at 1 (WA1/2020, $p$ = <0.0001; Nsp1-K164A/H165A, $p$ = 0.0001), 2 (WA1/2020 only, $p$ = 0.0002), 3 ($p$ < 0.0001), and 4 ($p$ < 0.0001) DPC by BA.1. By 5 DPC, infectious virus titers declined to baseline in all BA.1 challenged animals (Fig. 3c). At 4 DPC, subgenomic viral RNA (sgRNA) of the envelope (E) protein, a hallmark of viral replication, was readily detectable in the lungs, trachea, and nasal turbinates of Delta- and BA.1- challenged unvaccinated animals (Fig. 3d). By contrast, sgRNA levels were reduced by more than 4400-fold ($p \leq 0.0001$) in the lungs, around 300-fold in trachea ($p$ < 0.0001), and 300-fold in nasal turbinates ($p$ < 0.0001) of Nsp1-K164A/H165A vaccinated and WA1/2020 convalescent animals after challenge with the Delta variant. As to BA.1 challenged animals, the reduction of sgRNA was greater than 100-fold in lungs ($p$ < 0.0001), trachea ($p$ = 0.0041), and nasal turbinates ($p$ < 0.0001) at 4 DPC in both Nsp1-K164A/H165A vaccinated and WA1/2020 convalescent animals compared to the unvaccinated group. Infectious viral titers in lung homogenates also diminished below the limit of detection in Nsp1-K164A/H165A vaccinated and WA1/2020 convalescent groups following either Delta or Omicron BA.1 challenge at 4 DPC (Fig. 3e). At 7 DPC, sgRNA levels in the lungs were below the limit of detection except for the unvaccinated/challenged hamsters (Fig. 3f). Taken together, these

results indicate that intranasal vaccination with Nsp1-K164A/H165A effectively reduces viral loads in both upper and lower respiratory tract of Syrian hamsters upon heterologous virus challenge.

To further examine the presence of viral antigens and host innate immune activation, lung sections from uninfected (mock) or challenged hamsters were stained with hematoxylin and eosin (H&E) or immunostained for viral nucleocapsid protein (NP) and myxovirus resistance 1 (MX1), an interferon-induced antiviral host response marker[24,25] (Fig. 4). Lungs from Delta-challenged unvaccinated hamsters at 4 DPC showed widespread immune infiltrates and regions of viral NP deposition characterized by prominent staining of the epithelial lining of infected bronchioles accompanied by intense staining of surrounding alveolar epithelium (Fig. 4a, b). Two of the four BA.1-challenged unvaccinated animals at 4 DPC showed NP deposition with a similar staining pattern (Fig. 4b, c). Delta- and BA.1-challenged unvaccinated groups also showed increased MX1 immunoreactivity in these NP-positive lung regions, which is particularly evident in the bronchiolar epithelium. Vaccination with Nsp1-K164A/H165A or infection with WA1/2020 blocked NP deposition and MX1 upregulation in the Delta- and BA.1-challenged groups (Fig. 4b, c). High resolution imaging of representative lung sections from Delta-challenged unvaccinated hamsters highlighted the pronounced upregulation of MX1 in nuclear and cytoplasmic compartments of infected bronchiolar

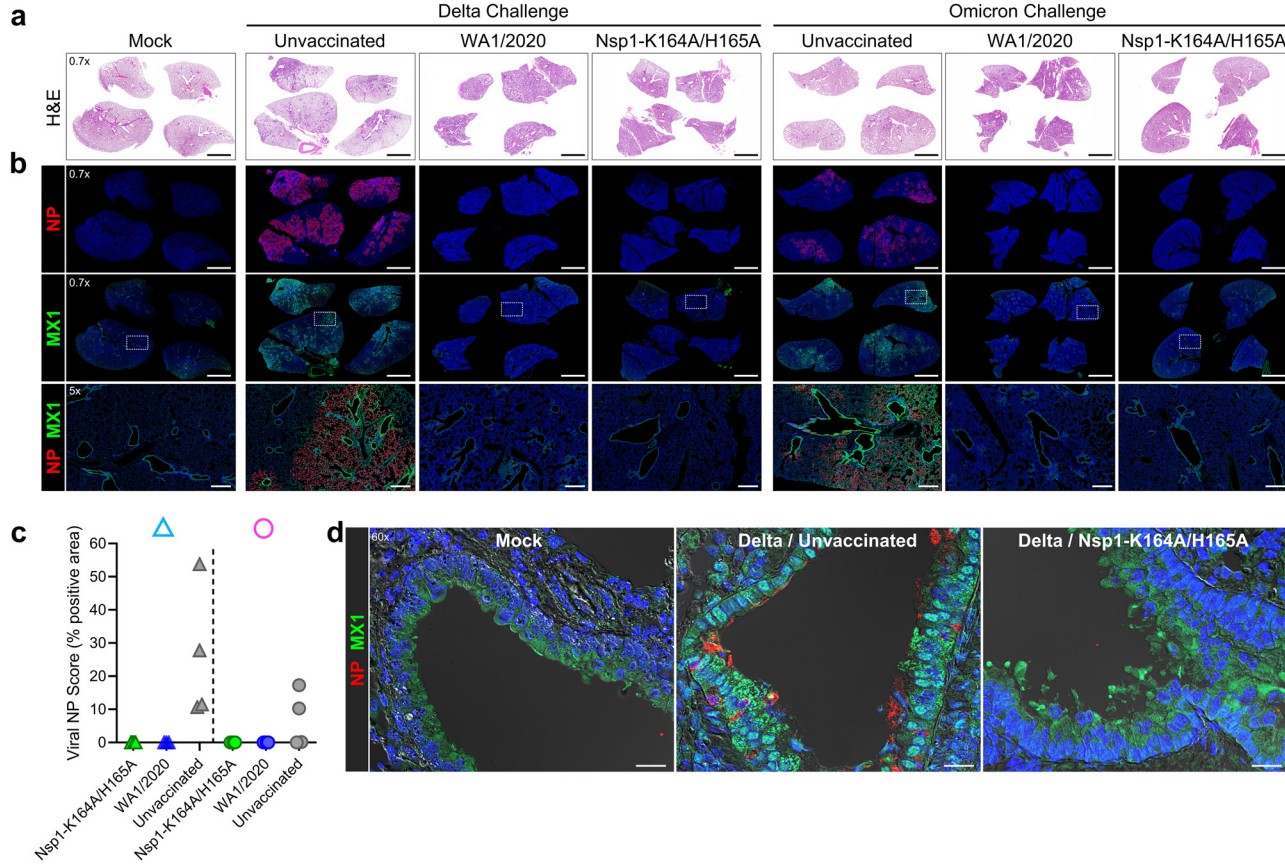

**Fig. 4 | Nsp1-K164A/H165A vaccination blocks virus propagation and MX1 induction in hamster lungs.** Syrian hamsters were vaccinated with a low (100 PFU) dose of Nsp1-K164A/H165A or WA1/2020 35 days prior to challenge with Delta or BA.1 isolate on day 0. Serial lung sections from non-infected non-vaccinated hamsters (mock) or hamster at 4 DPC were stained by **a** H&E or double-immunostained for **b** SARS-CoV-2 nucleocapsid protein (NP) and MX1 (interferon-induced antiviral protein). In **a**, images are shown at one level of magnification (×0.7) while corresponding serial immunostained images in **b** are shown at two levels of magnification (×0.7 and ×5) with white boxes delimiting the regions of magnification. **c** Semiquantitative analysis of viral NP staining in hamster lungs at 4 DPC. The plotted values represent the percent NP positive area as a function of the

total lung area for each section ($n$ = 3–4 animals per group). **d** High magnification immunofluorescence/differential interference contrast images of NP and MX1 in representative bronchioles of lung sections from mock hamsters or Delta-infected non-vaccinated or Nsp1-K164A/H165A-vaccinated hamsters at 4 DPC. Prominent cytoplasmic and nuclear localization of MX1 was detected in NP-positive bronchiolar epithelium in Delta-infected unvaccinated hamsters compared to low cytoplasmic expression of MX1 in mock and vaccinated hamsters. Nuclei were counterstained with Hoechst 33342 dye (blue). Scale bars: 5 mm (×0.7), 500 μm (×5), 20 μm (×60). Δ delta variant challenge, O omicron BA.1 challenge, DPC days post-challenge.

epithelial cells and the attenuation in Nsp1-K164A/H165A-vaccinated hamsters (Fig. 4d). Altogether, the absence of NP staining and MX1 upregulation implies that the challenge virus, whether it is the Delta or Omicron BA.1 variant, failed to establish infection in the lungs of Nsp1-K164A/H165A-vaccinated and WA1/2020-convalescent hamsters.

Among the Delta-challenged hamsters, only the unvaccinated group ($n = 8$) significantly lost body weight over the course of 7 days (Fig. 5a). By contrast, none of the three groups lost weight after BA.1 challenge (Fig. 5a). At 4 DPC, percent of lung consolidation in Delta variant-challenged hamsters was significantly reduced in WA1/2020- ($p = 0.0352$, $n = 3$) and Nsp1-K164A/H165A- inoculated groups ($p = 0.0386$, $n = 4$) compared to unvaccinated controls (Fig. 5b). Accumulated pathology scores were also significantly lower at 4 DPC in Delta-challenged WA1/2020 ($p = 0.0002$) and Nsp1-K164A/H165A groups ($p < 0.0001$) as opposed to the unvaccinated- and Delta-

challenged animals (Fig. 5b, c). Differing from the Delta variant, Omicron BA.1 challenge resulted in low pathology overall, with minimal consolidation (Fig. 5b) and low pathology scores at 4 DPC (Fig. 5c, d). At 7 DPC, among Delta-challenged hamsters, both Nsp1-K164A/H165A and WA1/2020 inoculated groups ($n = 3$) had significantly ($p < 0.0001$) reduced consolidation compared to unvaccinated controls (Fig. 5e). Following BA.1 challenge, only two animals in the unvaccinated group had >20% lung consolidation at 7 DPC (Fig. 5e), however, lung pathology was evident in 3 out of 4 animals within this group (Fig. 5f, g). Again, nearly no lung pathology was detected from Nsp1-K164A/H165A-vaccinated animals ($n = 3$, $p = 0.0195$).

To further characterize these histopathological changes using specific markers of inflammation and epithelial damage, serial lung sections from uninfected hamsters (mock), unvaccinated hamsters, and WA1/2020-convalescent and Nsp1-K164A/H165A-vaccinated

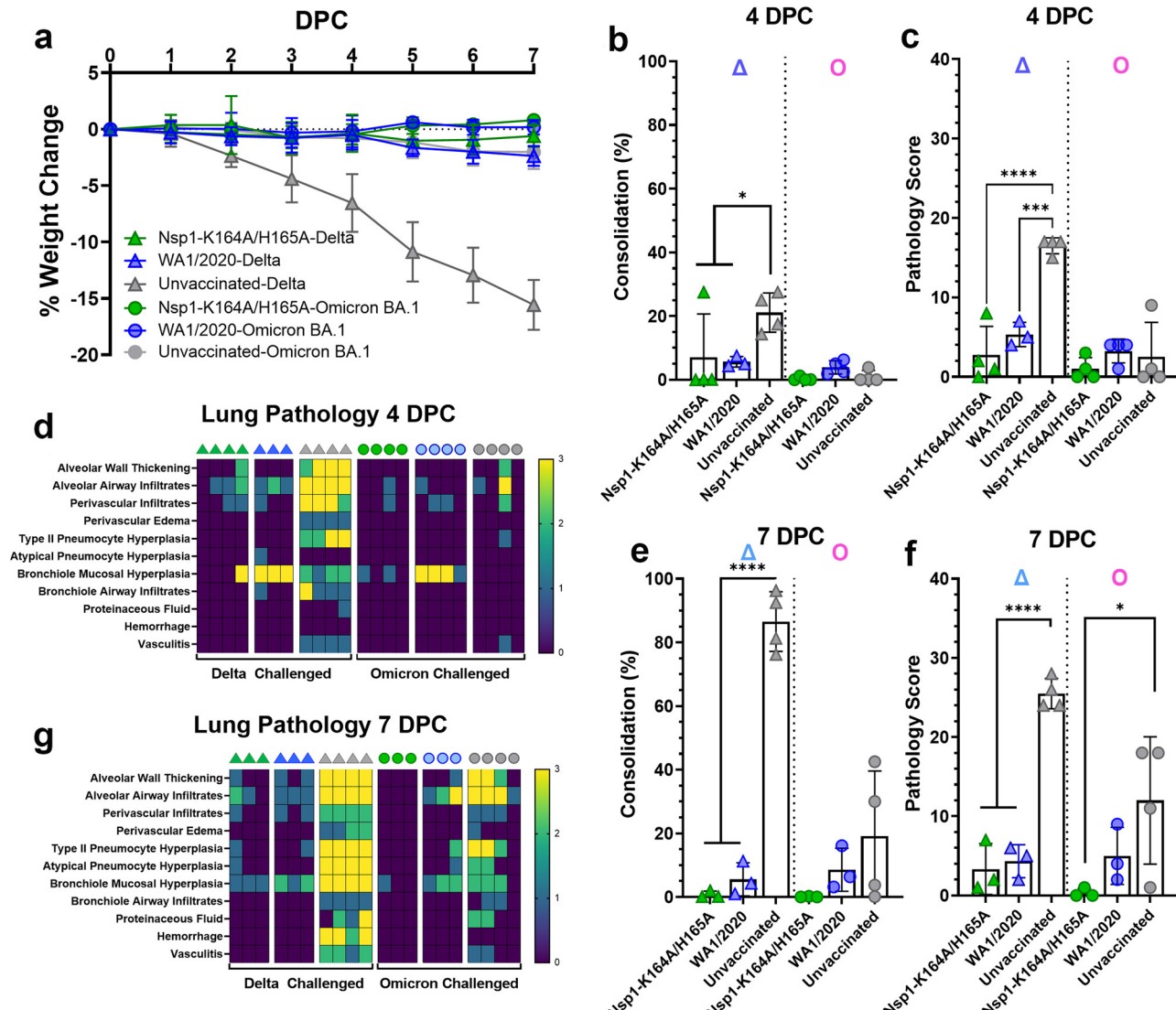

**Fig. 5 | Intranasal immunization of Syrian hamsters with Nsp1-K164A/H165A protects against Delta and Omicron challenge. a** Weight change was recorded for hamsters (described in Fig. 3a) after challenge by Delta and Omicron BA.1 variants for 7 days with points representing mean values with standard deviations indicated by error bars. Percentage of consolidation (*$p = 0.0352$ left, 0.0386 right) (**b**) and pathology score (***$p = 0.0002$) (**c**) in fixed lung tissues were compared between WA1/2020 convalescent ($n = 7$), Nsp1-K164A/H165A vaccinated ($n = 8$), and unvaccinated control ($n = 8$) lungs at 4 DPC. Individual pathologies were graded by severity and presented in a heat map (**d**). Percentage of consolidation (**e**) and

pathology score (**f**) were also compared at 7 DPC between WA1/2020 convalescent ($n = 6$), Nsp1-K164A/H165A vaccinated ($n = 6$), and unvaccinated control ($n = 8$) lungs (*$p = 0.0195$). **g** Heat-map presentation of individual pathologies at 7 DPC. Dot plots represent samples collected from individual animals in a single experiment with bars and error bars indicating mean values with standard deviations. Statistical differences were calculated using ordinary one-way analysis of variance (ANOVA) in GraphPad Prism 9.4.0 with Tukey's multiple comparisons tests. For statistical significance, *$p < 0.05$, ***$p < 0.001$, and ****$p < 0.0001$. Δ delta variant challenge, O omicron BA.1 challenge, DPC days post-challenge.

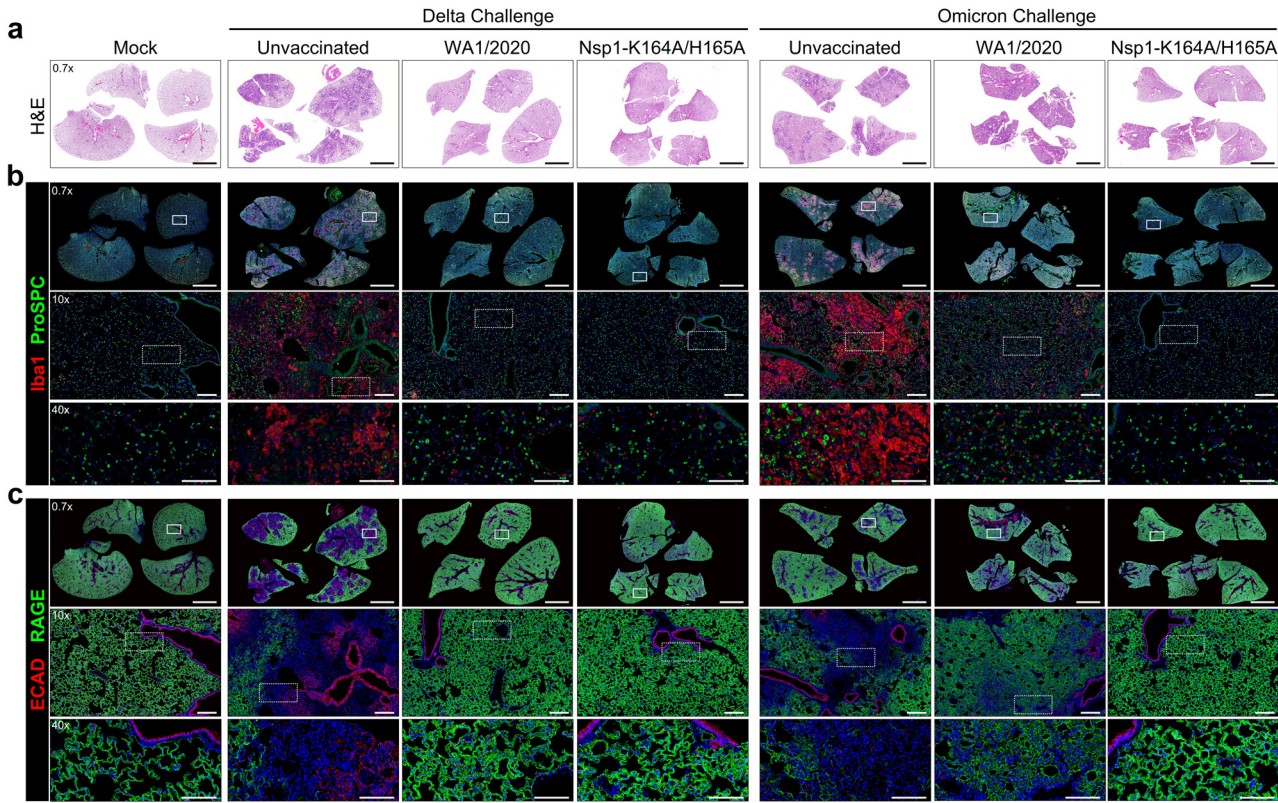

**Fig. 6 | Nsp1-K164A/H165A vaccination protects against lung pathology post-challenge with Delta and Omicron isolates.** Syrian hamsters were vaccinated with a low (100 PFU) dose of Nsp1-K164A/H165A or WA1/2020 35 days prior to challenge with Delta or BA.1 Omicron isolates on day 0. Serial lung sections from non-infected non-vaccinated hamsters (mock) or 7 DPC-infected hamsters were stained by **a** H&E or double-immunostained for either **b** Iba1 (macrophage marker) and Prosurfactant protein C (ProSPC, AT2 marker) or **c** E-cadherin (ECAD, epithelial junctional marker) and RAGE (AT1 marker). Delta-infected unvaccinated lungs ($n = 4$) show extensive areas of tissue consolidation (**a**) that correspond with regions showing abundant Iba1-labeled macrophage accumulation and loss of ProSPC-labeled AT2 cells (**b**) as well as loss of alveolar wall RAGE-expressing AT1 epithelium surrounding affected ECAD-stained bronchioles and aberrant reepithelization (**c**). Similar though less extensive pathology was observed in two of four BA.1-challenged unvaccinated hamsters. Nsp1-K164A/H165A or WA1/2020 inoculation completely prevented or suppressed Delta or BA.1-induced lung pathology. In **a**, images are shown at one level of magnification (×0.7) while corresponding serial immunostained images in **b** and **c** are shown at three levels of magnification (×0.7, ×10, and ×40) with white boxes delimiting the regions of magnification. Nuclei were counterstained with Hoechst 33342 dye (blue). Scale bars: 5 mm (×0.7), 250 μm (×10), 100 μm (×40).

hamsters were immunostained for Iba1 (a marker of macrophages), prosurfactant protein C (ProSPC, a marker of AT2 cells), RAGE (a marker of AT1 cells), and E-cadherin (a marker of intercellular epithelial junctions). At 7 DPC, lungs from all four Delta-challenged unvaccinated hamsters and two of the four BA.1-challenged unvaccinated hamsters showed regions of consolidation by routine H&E that corresponded with areas containing extensive accumulation of Iba1-expressing macrophages (Fig. 6a, b) and increased terminal deoxynucleotidyl transferase dUTP nick end labeling (TUNEL; Fig. S1). These consolidated regions also showed marked reduction of ProSPC-expressing AT2 cells and pronounced loss of alveolar RAGE expression along the borders of AT1 cells. Distinct areas of epithelial cell loss and Iba1-positive macrophage consolidation formed around E-cadherin-labeled bronchioles that were presumably subject to virus attack (Fig. 6c). Increased E-cadherin staining also identified hyperplastic epithelium in consolidated spaces that was particularly prominent in Delta-challenged unvaccinated lungs. Vaccination with Nsp1-K164A/H165A attenuated Iba1-positive macrophage consolidation and protected against the loss of ProSPC and RAGE expression in Delta- and BA.1-challenge groups (Fig. 6a–c). Prior infection with WA1/2020 also protected against epithelial damage in the Delta-challenge group.

### Transmission of Nsp1-K164A/H165A in Syrian Hamsters

To characterize the transmissibility of Nsp1-K164A/H165A, we performed an additional study in an airborne transmission model (Fig. 7a) in which two Syrian hamsters in the same cage are separated from each other by a customized, perforated metal divider that prevents physical contact while permitting air exchange (Supplementary Movie 1).

We first inoculated donor hamsters (male, 4-month-old, $n = 14$) with 100 PFU of WA1/2020 or Nsp1-K164A/H165A, which was previously shown to be immunogenic and protective against WA1/2020 challenge. Significant body weight loss was observed in WA1/2020 infected hamsters at days 3–9 ($p < 0.0001$), 10–11 ($p < 0.001$), and 12–14 ($p < 0.01$) DPI and one had to be euthanized due to presentation of severe clinical signs (hypothermia, hunched posture, lethargy) at 7 DPI (Fig. 7b). By contrast, no significant weight loss was observed in Nsp1-K164A/H165A vaccinated animals ($n = 14$). WA1/2020 inoculated hamsters also exhibited significantly higher infectious viral loads in nasal wash samples at 1, 2, 3 DPI than Nsp1-K164A/H165A inoculated group (1 DPI, 158-fold, $p < 0.0001$; 2 DPI, ~33-fold, $p = 0.0003$; and 3 DPI, 10-fold, $p = 0.0105$; Fig. 7c).

At 1 DPI, 7 sentinel hamsters were paired individually with either a WA1/2020 or a Nsp1-K164A/H165A inoculated hamster in cages with dividers. Nasal washes were collected from the sentinel hamsters from 1–4 days post-exposure (DPE) and seroconversion was determined after two weeks to confirm infection. Nasal wash infectious virus titers from Nsp1-K164A/H165A-exposed sentinels were undetectable at 1 DPE and then significantly lower at 2 DPE (>100-fold, $p = 0.0006$) and 3 DPE (>290-fold, $p < 0.0001$) compared to those from WA1/2020-exposed sentinels (Fig. 7d). At 4 DPE, there was no statistically significant difference between sentinel groups although 2 of the Nsp1-K164A/H165A sentinels had nasal wash titers below the detection limit (200 TCID$_{50}$/

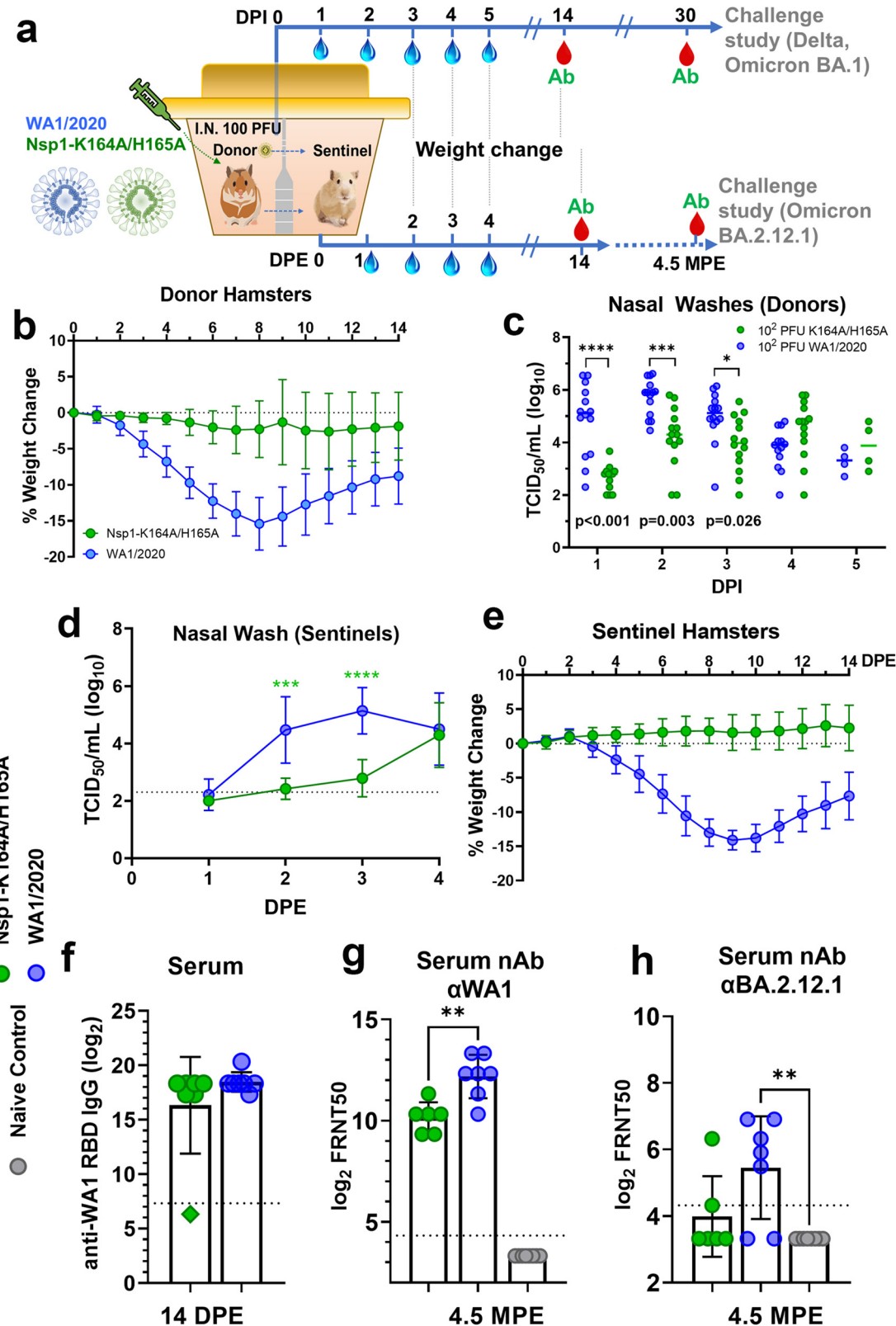

mL). Overall, the transmission of Nsp1-K164A/H165A to sentinel hamsters exhibited a delayed kinetics. No weight loss was evident in the sentinel hamsters exposed to Nsp1-K164A/H165A, however, WA1/2020-exposed sentinels experienced weight loss at days 4–5 ($p < 0.05$), 6 ($p = 0.0005$), 7–12 ($p < 0.0001$), 13 ($p = 0.0003$), and 14 ($p = 0.0016$) DPE with a maximum mean weight loss of 15.6% on 9 DPE (Fig. 7e). By 14 DPE all exposed sentinel animals had high levels of anti-RBD IgG in

serum (Fig. 7f) except one animal from the Nsp1-K164A/H165A-exposed sentinel group (animal ID WH363), which did not seroconvert (there was also no detectable infectious virus in nasal washes collected from this animal).

At 4.5 months after exposure (MPE), all seroconverted sentinel animals were tested for nAb titers against WA1/2020 and BA.2.12.1. Serum nAb titers against WA1/2020 from WA1/2020 exposed sentinel

**Fig. 7 | Airborne transmission of Nsp1-K164A/H165A in Syrian hamsters.**
**a** Donor Syrian hamsters (male, 5-month-old) were first inoculated with 100 PFU Nsp1-K164A/H165A ($n = 14$) or WA1/2020 ($n = 14$). One day after inoculation, donor hamsters were paired with recipient hamsters (sentinel, $n = 7$/group) in the specially designed cages with metal dividers for monitoring airborne transmission. During pairing, nasal swabs were collected daily from sentinel hamsters for 4 days. **b** Weight loss profile of donor hamsters ($n = 14$) after virus inoculation. **c** Infectious virus titers of nasal wash samples collected from donor hamsters were measured by a $TCID_{50}$ assay for up to 5 days post-inoculation and were compared using two-way ANOVA with Sidak's multiple comparison test. *$p = 0.0264$, ***$p = 0.0003$. Symbols indicate individual hamsters from a single experiment ($n = 14$ per group). Statistical differences were calculated by Student's unpaired $t$-test in GraphPad Prism 9.4.0. **d** Infectious virus titers of nasal wash samples collected from sentinel hamsters were measured by a TCID50 assay for up to 4 days post-exposure (DPE; $n = 7$ per group). ***$p = 0.0004$. **e** Weight loss profile of sentinel hamsters ($n = 7$ per group)

up to 14 days post-exposure to donor hamsters. **f** Seroconversion of sentinel hamsters was confirmed by ELISA measuring anti-WA1/2020 RBD IgG in Nsp1-K164A/H165A ($n = 7$) and WA1/2020 ($n = 7$) hamsters in seven independent pairs each. Note, one hamster (indicated by a green diamond) after exposure to Nsp1-K164A/H165A did not seroconvert. Serum nAB titers against WA1/2020 (**$p = 0.0026$) (**g**) and BA.2.12.1 (**h**) in sentinel hamsters ($n = 6$ for sentinels exposed to Nsp1-K164A/H165A and $n = 7$ for those exposed to WA/2020) after four and half months post-exposure (MPE) were measured by focus forming reduction neutralization assays (**$p = 0.0026$). Symbols in **b**, **e** indicate mean percent weight change data of groups of hamsters relative to initial individual animal weights on day 0 with standard deviation indicated by error bars. Bar graphs indicate mean values with error bars signifying standard deviations. Statistical differences were calculated in GraphPad Prism 9.4.0 using a two-way analysis of variance (ANOVA) with Sidak's multiple comparisons test or Student's unpaired $t$ test (**g**).

animals (GMT 4637, IQR 7680) were 4-fold higher ($p = 0.0026$, unpaired $t$ test) than Nsp1-K164A/H165A-exposed sentinel hamsters (GMT 1140, IQR 960; Fig. 7g). Neutralization of BA.2.12.1, however, was observed in only 2 animals from the Nsp1-K164A/H165A sentinel group and 5 animals from the WA1/2020 sentinel group prior to challenge (Fig. 7h).

### Passively vaccinated sentinel hamsters are protected from BA.2.12.1 challenge
Four and half months after the initial exposure, seroconverted sentinel hamsters from Fig. 7 were challenged with $10^4$ FFU Omicron BA.2.12.1 by the intranasal route. Weight loss did not occur in any BA.2.12.1 challenged hamsters (Fig. 8a). Reduced nasal viral loads in WA1/2020-exposed sentinel hamsters ($n = 7$) were noticed on days 3 ($p = 0.0492$) and 4 ($p = 0.0.0011$) DPC compared to naïve hamsters that were challenged with BA.2.12.1 (Fig. 8b). Interestingly, despite low or absent BA.2.12.1-specific nAb titers in Nsp1-K164A/H165A sentinel animals (Fig. 7h), nasal viral load of these animals was significantly lower compared to controls at 4 DPC ($p = 0.0094$; Fig. 8b). Infectious virus titers in BALF were at least one-log lower in WA1/2020 and Nsp1-K164A/H165A sentinel hamsters than in naïve controls at 4 DPC ($p < 0.0001$; Fig. 8c). Infectious virus titers in lung homogenates of the BA.2.12.1 infected control animals ($n = 4$) were around 1000 FFU/mL at 4 DPC, whereas 2 out 4 sentinel hamsters had no detectable infectious virus in the lungs (Fig. 8d). Consequently, $log_{10}$-transformed sgRNA copies detected in both Nsp1-K164A/H165A ($p = 0.0043$) and WA1/2020 sentinel hamster lungs ($p = 0.0004$) were at least one order of magnitude lower compared to controls at 4 DPC (Fig. 8e).

Histopathology analyses of fixed lungs revealed minimal consolidation at 4 DPC ($n = 4$) and 7 DPC ($n = 3$) in all groups. A slightly higher percentage of lung consolidation (Fig. 8f) was observed in WA1/2020-exposed sentinels compared to Nsp1-K164A/H165A (4 DPC, $p = 0.0490$) or Nsp1-K164A/H165A sentinel hamsters ($p = 0.0269$) and control animals ($p = 0.0326$) at 7 DPC. The consolidation is most likely remnant pathology from the initial WA1/2020 infection, but not due to BA.2.12.1 challenge. There was no significant difference in pathology scores at 4 or 7 DPC between groups (Fig. 8g), although WA1/2020-exposed sentinel hamsters did show bronchiole mucosal hyperplasia at 4 DPC (Fig. 8h) and 7 DPC (Fig. 8i). Noticeable lung pathologies in naive animals after BA.2.12.1 challenge included alveolar wall thickening, airway infiltrates, and type II pneumocyte hyperplasia. By contrast, such pathologies were absent in the lungs of Nsp1-K164A/H165A exposed sentinel hamsters after challenge (Fig. 8h, i). Altogether, Nsp1-K164A/H165A exposed sentinel hamsters were protected from BA.2.12.1 challenge in the lungs.

### Nsp1-K164A/H165A vaccination protects against BA.5 challenge
Because Omicron BA.1 and BA.2.12.1 only caused mild diseases in Syrian hamsters, we performed another vaccination-challenge study using

a BA.5 isolate that induces more severe disease. Shown in Fig. 9a, sixty days after vaccination with 100 PFU Nsp1-K164A/H165A, Syrian hamsters were challenged with $10^4$ PFU BA.5 isolate. Unvaccinated animals rapidly lost weight over the next 7 days, whereas the body weight of vaccinated animals remained steady throughout the course of the study (Fig. 9b). Vaccinated animals had detectable virus from nasal wash samples collected at 1 DPC with minimal (at or below) infectious virus at subsequent timepoints. By contrast, unvaccinated animals shed at least 2 logs higher infectious virus in nasal wash samples from 1 to 3 DPC (Fig. 9c). Viral loads in nasal turbinates, BALF, and lungs, were nearly undetectable in vaccinated animals at 4 and 7 DPI (Fig. 9d–g). Lastly, Nsp1-K164A/H165A vaccinated hamsters also showed very little lung pathology at 4 and 7 DPC in comparison to unvaccinated animals (Fig. 9h–k). It is worth mentioning that BA.4/BA.5 and some later Omicron subvariants are even more immune evasive than the initial BA.1 and BA.2 variants, but our prototype Nsp1-K164A/H165A LAV conferred protection against BA.5 both in the upper and lower respiratory tract in Syrian hamsters.

### Variant-specific Nsp1-K164A/H165A as a booster
To test if Nsp1-K164A/H165A may be used as a booster vaccine candidate, we generated two additional attenuated viruses with the WA1 spike being replaced with either BA.1 or BA.5 spike protein, namely BA.1-LAV and BA.5-LAV, respectively (Fig. 10a). Notably, we have previously published that BA.1 enters cells expressing mouse ACE2 (mACE2) and infects laboratory mouse strain such as BALB/c[26]. In this study, we also confirmed that BA.5 infects 293T cells expressing mACE2 (Fig. 10b). Hence, we envisioned that BA.1-LAV and BA.5-LAV would infect BALB/c and induce variant-specific antibody response. To test this possibility, BALB/c mice first received two doses of vaccinia virus Ankara vectors expressing full-length WA1 Spike (MVA-S)[27]. After 8 weeks, $10^4$ PFU BA.1-LAV or BA.5-LAV were intranasally administered to these mice (Fig. 10c)[26]. A dose of $10^4$ PFU was chosen to ensure that the candidate vaccine virus infects the mice with pre-existing immunity. It is noted that pre-boost sera contained high levels of nAbs against the original WA1 isolate, but nondetectable anti-BA.1 or anti-BA.5 nAb. Importantly, two weeks after one dose of BA.1-LAV and BA.5-LAV, the GMT values of corresponding nAb titers increased to 143 (IQR 387.6) and 84 (IQR 315.9), respectively (Fig. 10d).

## Discussion
Current SARS-CoV-2 vaccines, particularly those based on mRNA, induce robust systemic humoral and cellular immunity, and prevent severe disease caused by SARS-CoV-2[28]. However, protection against infection and transmission of SARS-CoV-2 variant viruses, in particular the Omicron sub-lineage viruses, by mRNA vaccines may be limited[29–31]. Results from recent studies suggest that intramuscular vaccination tends to induce potent systemic but not local humoral response at the mucosa. By contrast, nasal vaccines induce more

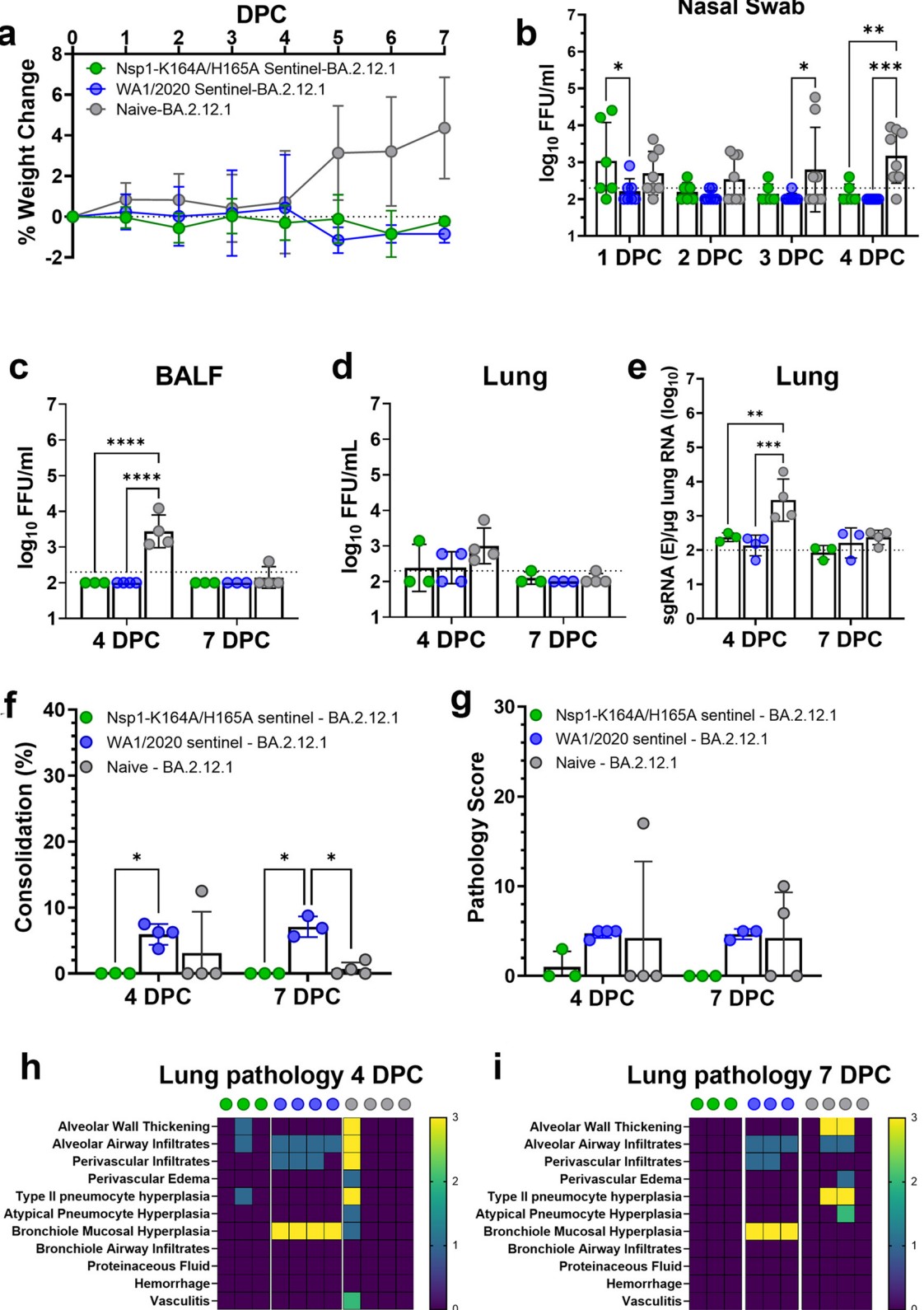

robust mucosal immune responses, characterized by secretion of IgA and IgG at mucosal surfaces and detection of resident memory T/B cells in respiratory tract[16,23,32–34]. Here we show that intranasal immunization of Syrian hamsters with a live attenuated SARS-CoV-2 (WA1-ΔPRRA-ORF6-8-Nsp1[K165A/H165A]) induced SIgA and IgG at respiratory mucosa, albeit the levels of mucosal IgA/IgG appeared to be at least two orders of magnitude lower than their serum counterparts.

Together, our study demonstrates the great potential of live attenuated WA1-ΔPRRA-ORF6-8-Nsp1[K164A/H165A] as a nasal vaccine capable of inducing SIgA/IgG in the respiratory mucosa.

Many of the pathologies found in human cases of COVID-19 are recapitulated in mature (>4–6-month-old) Syrian hamsters, making them an excellent animal model for studying SARS-CoV-2 pathogenesis[35]. For example, the damage to AT1 and AT2 cells with

**Fig. 8 | Seroconverted sentinel hamsters (i.e. passively vaccinated through transmission of Nsp1-K164A/H165A) are protected from BA.2.12.1 challenge.**
**a** Seroconverted sentinel hamsters (4.5 months after exposure to WA1-2020 and Nsp1- K164A/H165A) were challenged with $10^4$ PFU of BA.2.12.1. A group of 8 age-matched naïve hamsters were also included in the challenge study as controls. Weight change was recorded for 7 days post-challenge. **b**–**d** Infectious virus titers from nasal swabs (*$p = 0.0393$, *$p = 0.0416$, **$p = 0.0059$, ***$p = 0.0008$) (**b**), BALF (**c**), and lung homogenates (**d**) were measured by focus-forming assays. **e** Viral sgRNA levels in the lungs were quantified by RT-pPCR 4 DPC and 7 DPC (**$p = 0.0043$, ***$p = 0.0004$). Dot plots represent samples collected from individual

animals in a single experiment. Percentage of consolidation (*$p = 0.0490$, *$p = 0.0326$, *$p = 0.0269$) (**f**) and pathology scores in lungs (**g**) of sentinel hamsters ($n = 6$ for Nsp1-K164A/H165A, $n = 7$ for WA1/2020, and $n = 8$ for naïve controls) at 4 and 7 DPC with $10^4$ PFU of BA.2.12.1. Individual lung pathologies at 4 DPC (**h**) and 7 DPC (**i**) are presented in heat maps. Dot plots in this graph represent samples collected from individual animals in a single experiment. Statistical differences were calculated using ordinary two-way analysis of variance (ANOVA) in GraphPad Prism 9.4.0 with Tukey's multiple comparisons tests. For statistical significance, ****$p < 0.0001$. DPC days post-challenge.

the prominent infiltration of macrophages observed in the lungs of Delta- and BA.1-challenged unvaccinated hamsters partially mimics the histopathology of COVID-19 in humans[36,37]. One of the challenging aspects of this model, however, is the relative lack of species-specific reagents that has made immunological analyses in Syrian hamsters difficult. In this study we established methods to reliably measure SIgA/IgG in nasal wash and BALF samples. These techniques have now enabled us to assess the mucosal immunogenicity of SARS-CoV-2 vaccine candidates in Syrian hamsters. We detected significant amounts of SIgA/IgG in nasal wash and BALF samples from hamsters after intranasal vaccination of WA1-ΔPRRA-ORF6-8-Nsp1[K164A/H165A]. One noteworthy observation is that when SIgA/IgG is specifically raised against the ancestral virus (WA1/2020) and does not efficiently neutralize Omicron subvariants, vaccinated hamsters were not free of virus in the upper respiratory tract upon challenge with Omicron sub-lineage viruses. This finding is in line with the recent observation that anti-Spike mucosal IgA protects against SARS-CoV-2 Omicron infection in human population[38]. The durability of the mucosal IgA response is currently under investigation. Interestingly, Nsp1-K164A/H165A-vaccinated animals were protected in the lung against Omicron BA.1, BA.2.12.1, and even BA.5 challenge despite very little detectable nAbs in these animals against these newer variants. It is possible that cellular immunity, induced by Nsp1-K164A/H165A, contributes to the protection against heterologous virus challenge, as we detected splenocytes reacting to the nucleocapsid protein of the virus. Future research is warranted to decipher the mechanism of attenuated viral vaccines in conferring cross-protection in hamsters as more reagents become available.

The efficacy of our LAV candidate is on par with that of natural immunity acquired from previous infection, at least in Syrian hamsters[22,39,40]. An earlier study has shown that immunity acquired from previous infection was more protective than mRNA vaccination against re-infection with the BA.1 variant in hamsters[40]. In consistence, our prototype Nsp1-K164A/H165A-vaccinated animals were largely protected against BA.5 challenge in both upper and lower respiratory tract. In case the composition of the LAV candidate requires an update, however, we were able to rapidly construct attenuated SARS-CoV-2 bearing BA.1 and BA.5 spike protein using our published attenuation strategy[22]. Most intriguingly, attenuated SARS-CoV-2 bearing BA.1 and BA.5 spike readily boosted the neutralizing antibody titers against BA.1 and BA.5, respectively, in mice that were previously vaccinated with a MVA vector expressing the spike protein of the ancestral WA1/2020 isolate.

An early and robust activation of interferon (IFN) signaling pathways contributes to the protective mucosal immune response against viral infections in the respiratory tract[41]. Viral recognition in infected immune and/or epithelial cells triggers the production of IFNs (types I, II, and III) that subsequently limit viral replication and dissemination by activating the transcription of numerous antiviral IFN-stimulated genes (ISGs) including IFIT1-3, OAS1, IRF7, and MX1 in infected cells and bystander cells. While IFNs are critical to the antiviral host defenses, excessive or persistent IFN signaling may also aggravate lung pathologies of viral infections including SARS-CoV-2[42]. Activation of IFN signaling in the lungs of Delta- and BA.1 unvaccinated animals was

clearly evidenced by the robust upregulation of MX1 protein that localized specifically to infected bronchioles and neighboring alveolar parenchyma stained by viral NP. In this regard, we surmise that the lack of MX1 upregulation in the lungs of Nsp1-K164A/H165A-vaccinated hamsters supports that the vaccine exerts a strong and specific protective response along the upper airways that blocks viral dissemination in the lungs. The lack of macrophage accumulation in the lungs of Nsp1-K164A/H165A-vaccinated hamsters suggests that robust viral infection did not occur, hence vaccinated animals did not develop pneumonia.

An intrinsic safety concern for a LAV against a pandemic virus is the potential transmission of the vaccine virus. In principle, transmission of a vaccine virus may contribute to the establishment of herd immunity, but there is a possibility that an attenuated virus may directly or through reversion to a more virulent form cause illness among some at-risk individuals. We have previously demonstrated the genome stability of Nsp1-K164A/H165A[43]. Here we found that the airborne transmission of Nsp1-K164A/H165A was less efficient compared to wild-type SARS-CoV-2 in hamsters, but most (except for one animal) sentinel hamsters became seroconverted after 14 days. Airborne transmission of Nsp1-K164A/H165A did not cause weight loss in sentinel hamsters. Another intriguing finding is that four and half months after the initial exposure, these seroconverted hamsters were largely protected from a BA.2.12.1 challenge. Thus, passive immunization via transmission of Nsp1-K164A/H165A was achieved. Nonetheless, the risks and benefits of administering an infectious attenuated SARS-CoV-2 vaccine will need to be further evaluated.

## Study limitations
Typically, SARS-CoV-2 infection of Syrian hamsters induces pronounced consolidation and pathologies in the lung at 7 DPC[35]. However, lung pathology was limited to a subset of BA.1 and BA.2.12.1-infected hamsters rather than being widespread as in Delta challenged animals. For this reason, determination of vaccine efficacy against Omicron sub-lineage viruses using Syrian hamsters was problematic due to the low pathogenicity of BA.1 and BA.2.12.1 in this species. Low pathology scores in unvaccinated controls led to difficulties in determining efficacy of vaccination in preventing lung damage post-challenge. Additionally, persistent damage after WA1/2020 infection likely led to increases in bronchiole mucosal hyperplasia in hamsters observed at 4 DPC in convalescent animals challenged with Delta as well as BA.1 and BA.2.12.1 Omicron. In separate experiments being prepared for publication, we have observed long-term lung damage (including hyperplasia in bronchioles) in hamsters up to 4 weeks after WA1/2020 infection by the intranasal route. This would provide an explanation for the observed lung pathology in WA1/2020 convalescent hamsters after Omicron challenge which was not observed in control or Nsp1-K164A/H165A vaccinated animals.

WA1-ΔPRRA-ORF6-8-Nsp1[K164A/H165A] induces humoral immunity at mucosa as well as cellular-immunity targeting nucleocapsid protein. Transmission of WA1-ΔPRRA-ORF6-8-Nsp1[K164A/H165A] leads to immunization. Additionally, attenuated viruses bearing BA.1 and BA.5 spike

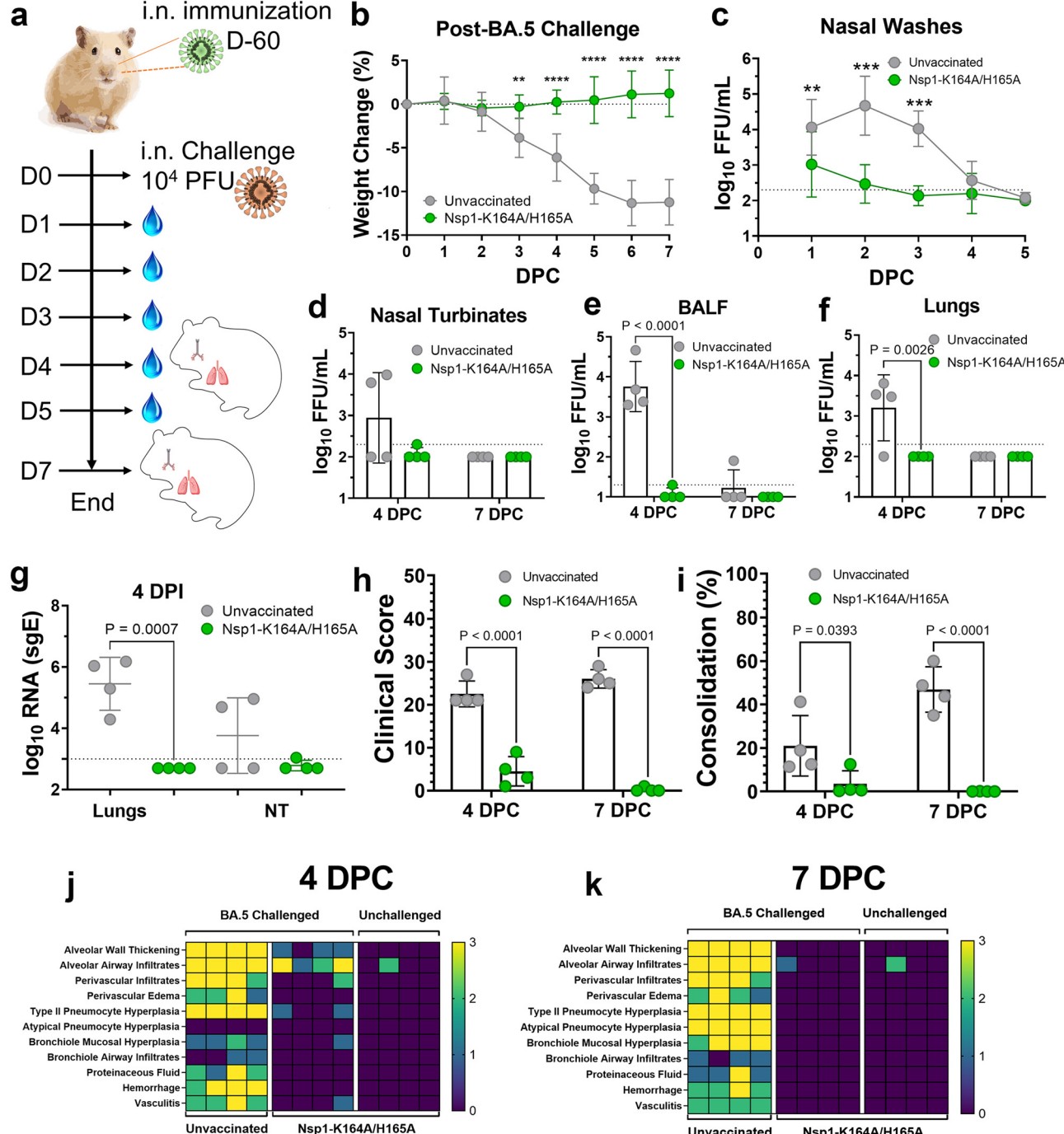

**Fig. 9 | Intranasal immunization of Syrian hamsters with 100 PFU Nsp1-K164A/H165A is protective against Omicron BA.5 challenge. a** Syrian hamsters (male, 5 months old) were vaccinated with 100 PFU Nsp1-K164A/H165A 60 days prior to challenge with $10^4$ PFU BA.5 (isolate hCoV-19/USA/COR-22-063113/2022) ($n=8$) on day 0. **b** Changes in weight were followed in Nsp1-K164A/H165A vaccinated and unvaccinated ($n=8$ each) challenged hamsters 0–7 DPC, with **$p=0.0030$ and ****$p<0.0001$. **c** from 1 to 5 DPC, infectious virus from nasal wash samples was quantified by focus-forming assays for vaccinated and unvaccinated hamsters ($n=4$) with **$p=0.0066$ and ****$p<0.0001$. **d–f** Infectious virus titers of nasal turbinates (d), bronchoalveolar lavage fluid (BALF, **e**), and lung homogenates (**f**) at 4 ($n=4$) and 7 ($n=4$) DPC were determined by focus-forming assays. **g** Viral sgRNA levels in lung and nasal turbinate samples from 4 DPC ($n=4$) were measured by qRT-PCR. Sum clinical scores (**h**) and percentage of consolidation (**i**) were also compared for lungs collected at 4 and 7 DPC. *P*-values are indicated in the graph where appropriate ($p<0.05$). Heat-map presentation of individual pathologies in lungs collected at 4 DPC (**j**) and 7 DPC (**k**). Graphs for **b** and **g** indicate mean values from a single experiment with standard deviations shown as error bars. Dot plots represent samples collected from individual animals in a single experiment, horizontal bars indicate mean values with standard deviations shown as error bars. Statistical differences were calculated using ordinary two-way analysis of variance (ANOVA) in GraphPad Prism 9.4.0 with Tukey's multiple comparisons tests.

boost variant-specific neutralizing antibodies in mice with pre-existing antibodies against the ancestral virus. Thus, WA1-ΔPRRA-ORF6-8-Nsp1[K164A/H165A] may be further developed into a nasal vaccine for primary series or as a booster.

## Methods

Research described here complies with all relevant ethical regulations and has been approved by the US Food and Drug Administration Institutional Biosafety Committee.

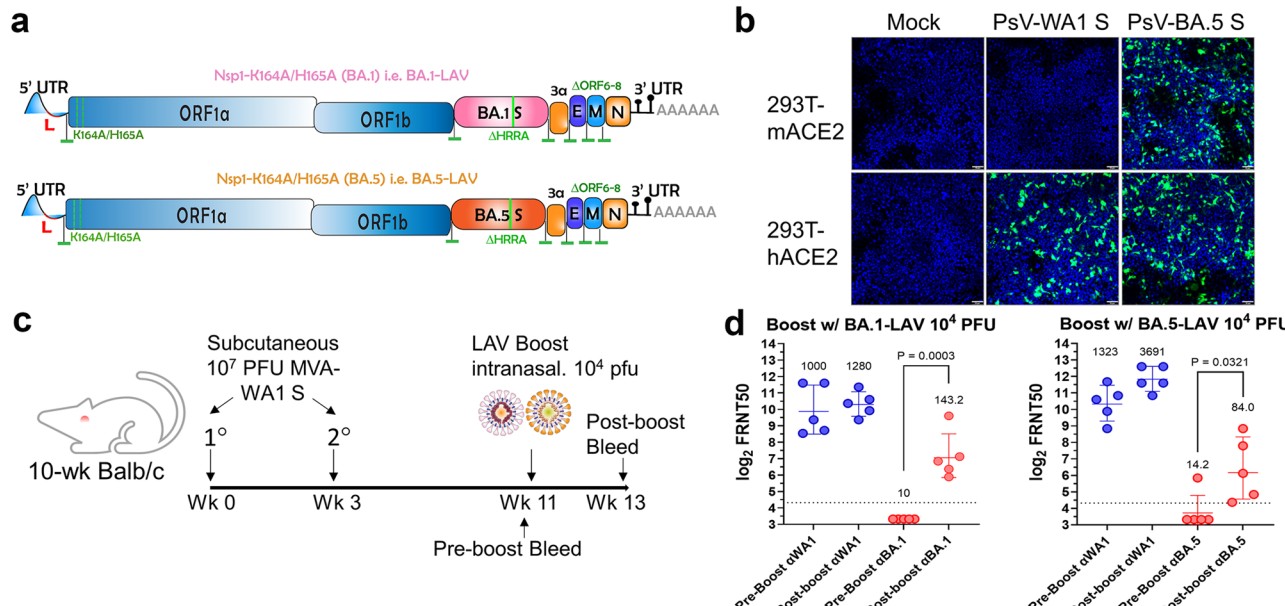

**Fig. 10 | BA.1-LAV and BA.5-LAV as boosters. a** Genome organization of BA.1-LAV and BA.5-LAV. Leader sequence (red), transcriptional regulatory sequence within the leader sequence and within the body are highlighted as green bars. The poly-basic insert "HRRA" was removed from the spike protein proteins and ORF6-8 were removed from the WA1/2020 backbone. Locations of K164A/H165A are highlighted in the figure. **b** Pseudovirus (PsV) bearing BA.5 spike but not WA1/2020 spike infected 293-mACE2 cell line. Infected cells are shown in green. Blue, DAPI strained nuclei. Scale bars in the lower right corner indicate 50 μm. **c** Overall study design to test BA.1-LAV and BA.5-LAV as boosters. 10-week-old male BALB/c mice were employed in this study. **d** Neutralizing antibody titers were measured from pre- and post-boost sera in BA.1 and BA.5 boosted mice (*n* = 5 in each group). Each solid circle represents one animal. Numbers above each group and average bars indicate the geometric means of neutralizing antibody titers with error bars signifying standard deviations. Values were compared using Student's unpaired *t* test in GraphPad Prism 9.4.0.

## Cells and viruses

Vero E6 cell line (Cat # CRL-1586) was purchased from American Type Culture Collection (ATCC) and cultured in Dulbecco's minimal essential medium (MEM) supplemented with 10% fetal bovine serum (Invitrogen) and 1% penicillin/streptomycin and L-glutamine. Calu-3 cell line (Cat # HTB-55) was obtained from ATCC and maintained in EMEM + 20%FBS. H1299-hACE2 is a human lung carcinoma cell line stably expressing human ACE2. The cell line was generated by lentiviral transduction of the NCI-1299 human lung carcinoma cell line (ATCC CRL-5803) with pLVX-hACE2 and selected with 1 μg/mL puromycin. A western blot was performed to confirm the expression of hACE2. H1299-hACE2 cells were maintained in DMEM supplemented with 5% penicillin and streptomycin, and 10% fetal bovine serum (FBS) at 37 °C with 5% $CO_2$.

The SARS-CoV-2 isolate WA1/2020 (NR-52281, lot 70033175) was obtained from BEI Resources, NIAID, NIH, and had been passed three times on Vero cells and 1 time on Vero E6 cells prior to acquisition. It was further passed once on Vero E6 cells in our lab. SARS-CoV-2 hCoV-19/USA/MD-HP05647/2021 (Delta variant, Pango lineage B.1.617.2) was obtained from BEI resources, NIAID, NIH (NR-55672, Lot 70046635) and had been passaged once in Vero E6-TMPRSS2 and once in Calu-3 cells prior to acquisition. It was passaged once more in H1299-hACE2 cells in our lab to generate viral stocks. Passaged viruses were deep sequenced to confirm identity (100% match with the original sequence, i.e., free of tissue culture adaptive mutations such as the loss of the polybasic site between S1 and S2 subunit of the spike protein). SARS-CoV-2 isolates hCoV-19/USA/HI-CDC-4359259-001/2021 (B.1.1.529 Omicron, NR-56475), hCoV-19/USA/NY-MSHSPSP-PV56475/2022 (BA.2.12.1 Omicron, NR-56782), USA/MD-HP30386/2022 (BA.4, Omicron, NR-56802), and hCoV-19/USA/COR-22-063113/2022 (BA.5 Omicron, NR-58620) were obtained from BEI resources and used directly in experiments. Recombinant SARS-CoV-2 viruses were generated as described previously[22,43]. The BA.1-LAV and BA.5 LAV were generated using standard molecular biology techniques. In brief, the WA1 spike sequence in the prototype Nsp1-K164A/H165A virus was replaced with corresponding BA.1 and BA.5 spike protein sequences. The polybasic insert "HRRA" was removed.

## Hamster challenge experiments

Adult male outbred Syrian hamsters were previously purchased from Envigo and held at the U.S. Food and Drug Administration (FDA) vivarium. All experiments were performed within the biosafety level 3 (BSL-3) suite on the FDA White Oak campus. The animals were implanted subcutaneously with IPTT-300 transponders (BMDS), randomized, and housed as compatible pairs (2 per cage) in sealed, individually ventilated rat cages (Allentown). Cages were maintained under controlled light (12:12 Light/Dark cycle), temperature (72 °F ± 1 °F), and humidity (30–70%) conditions. Hamsters were fed irradiated 5P76 (Lab Diet) *ad libitum*, housed on autoclaved aspen chip bedding with reverse osmosis-treated water provided in bottles, and all animals were acclimatized at the BSL3 facility for 4–6 days or more prior to the experiments. The study protocol details were approved by the White Oak Consolidated Animal Care and Use Committee and carried out in accordance with the PHS Policy on Humane Care & Use of Laboratory Animals (ASP 2020-06).

Adult male (5–6 months old) Syrian hamsters (*Mesocricetus auratus*) were anesthetized with (3-4% v/v) isoflurane and oxygen following procedures as described previously[35,44,45]. Intranasal inoculation was done by pipetting $10^2$ PFU or $10^4$ PFU SARS-CoV-2 in 50 μl volume dropwise into the nostrils of the hamster under anesthesia. Following infection, hamsters were monitored daily for clinical signs and weight loss. Nasal washes were collected by pipetting ~200 μl sterile phosphate buffered saline into one nostril when hamsters were anesthetized by 3–5% isoflurane. Nasal swabs were done as described previously[32].

For airborne transmission, a subset ($n = 7$) of hamsters inoculated with $10^2$ PFU WA1/2020 or Nsp1-K164A/H165A were paired in divided cages to prevent direct contact to measure transmission to naive sentinels[46]. One hamster (WH363), paired with an actively shedding Nsp1-K164A/H165A vaccinated animal, did not show evidence of productive infection or seroconvert at 14 DPE and remained seronegative until just prior to BA.2.12.1 challenge 4.5 months later. For these reasons, WH363 was removed from the challenge datasets.

For tissue collection, a subset of hamsters was humanely euthanized by intraperitoneal injection of pentobarbital at 200 mg/kg at 4 and 7 DPC. Lungs, trachea, and nasal turbinates were dissected for histopathology or homogenized for RNA extraction or titration in cell culture. Blood collection was performed under anesthesia (3–5% isoflurane) through gingival vein puncture or cardiac puncture when animals were euthanized. The left lobes of hamster lungs (~0.2 gram) were diced, divided, and resuspended in 1 milliliter MEM or TriZol reagent (RNA extraction) and homogenized on a Precellys Evolution tissue homogenizer with a Cooling Unit (Bertin). Trachea and nasal turbinates were homogenized the same way in TriZol Reagent. Splenocytes were extracted at 14 DPI from vaccinated and naive hamsters and IFNγ-secreting cells were identified after stimulation with spike and nucleocapsid antigen pools (BEI Catalog No. NR-52418 and NR-52419) by ELISpot (MABTECH, 3102-2H).

### RNA isolation and qRT-PCR
Procedures as described previously[35,44]. In brief, RNA was extracted from 0.1-gram tissue homogenates using QIAamp vRNA mini kit or the RNeasy 96 kit (QIAGEN) and eluted with 60 μl of water. 5 μL RNA was used for each reaction in real-time RT-PCR. When graphing the results in Prism 9, values below the limit of quantification (LoQ) were arbitrarily set to half of the LoQ values. Unless otherwise specified, the unit for RNA copies are as presented as $Log_{10}$ RNA copes/μg tissue RNA.

### Histopathology analyses
Procedures as described previously[35,44]. Tissues (lungs, trachea, and nasal turbinates) were fixed in 10% neutral buffered formalin overnight and then processed for paraffin embedding. The 5-μm sections were stained with hematoxylin and eosin for histopathological examinations. Images were scanned using an Aperio ImageScope. Blinded samples were graded by a licensed pathologist for the following twelve categories: consolidation, alveolar wall thickening, alveolar airway infiltrates, perivascular infiltrates, perivascular edema, type II pneumocyte hyperplasia, atypical pneumocyte hyperplasia, bronchiole mucosal hyperplasia, bronchiole airway infiltrates, proteinaceous fluid, hemorrhage, and vasculitis. Grading: $0 = $ none, $1 = $ mild, $2 = $ moderate, $3 = $ severe. A graph was prepared by summing up the score in each category.

### Virus titration
Tissue culture infectious dose 50% ($TCID_{50}$) assays were done described previously[35,44] for initial nasal wash titrations post-inoculation. In brief, Vero E6 cells were plated the day before infection into 96-well plates at $1.5 \times 10^4$ cells/well. On the day of the experiment, serial dilutions of 20 μl nasal wash samples were made in media and a total of six to eight wells were infected with each serial dilution of the virus. After 48 h incubation, cells were fixed in 4% PFA followed by staining with 0.1% crystal violet. The TCID50 was then calculated using the formula: log (TCID50) = log(do) + log (R) (f + 1). Where do represents the dilution giving a positive well, f is a number derived from the number of positive wells calculated by a moving average, and R is the dilution factor.

For focus-forming assay, nasal wash, BALF, and lung homogenate samples were 10-fold serially diluted in 96-well plates and dilutions added to 96-well black-well plates for fluorescent focus forming assays in H1299-hACE2 cells[47]. After 1 h the Tragacanth gum overlay (final concentration 0.3%) was added. Cells were incubated at 37 °C and 5% $CO_2$ for 1 day, then fixed with 4% paraformaldehyde, followed by staining of cells with primary rabbit anti-N Wuhan-1 antibody (Genscript) overnight followed by secondary anti-rabbit Alexa-488 conjugated antibody and DAPI staining. The infectious titers were then counted using Gen5 software on a Cytation7 machine and calculated and plotted as focus forming units per milliliter (FFU/ml). Limits of detection for the FFA were set based on the minimum detectable titer given 1 focus-forming unit at the lowest dilution ($10^{-1}$) and inoculation volume (50 μl). Any values below the lower limit (200 FFU/ml) were arbitrarily assessed as 100 FFU/ml for statistical analysis.

### SARS-CoV-2 neutralization assay
Samples were serially diluted 2-fold in 5% FBS DMEM and mixed with 100 PFU of SARS-CoV-2 in a 96-well plate at 37 °C for 1 h. Sample:virus mixtures were then added to confluent H1299-hACE2 cells in 96-well plates. Cells were infected for 1 h before the inoculum was removed and washed three times with DPBS. A second overlay containing 1.2% Tragacanth gum, 2X MEM, 5% FBS, and DMEM was added to the plate. Cells were incubated at 37 °C for 1 day, then fixed with 4% paraformaldehyde, followed by staining of cells with primary rabbit anti-SARS-CoV-2 N antibody (Genscript U739BGB150-5) overnight followed by secondary anti-rabbit Alexa-488 conjugated antibody and 4′,6-diamidino-2-phenylindole (DAPI) staining. Plates were imaged on a Cytation7 (Agilent), and foci were counted using Gen5 software. For the neutralization assays, recombinant LY-CoV555 (Bamlanivimab) mixed with WA1/2020[19] was included as a positive control. The 50% endpoint neutralization titers were determined as the reciprocal of the highest dilution providing ≤ half of the number of foci obtained from the negative control well (plain DMEM mixed with 100 PFU virus).

### Measurement of antibody by ELISA
The preparation of SARS-CoV-2 RBD antigen in a baculovirus expression system and its use in ELISA were previously described[27]. ELISAs were performed with slight modifications. Briefly, Immulon $2^{HB}$ plates were coated with recombinant RBD protein at 1 μg/mL overnight at 4 °C. Test serum samples were pre-diluted in assay diluent (PBS containing 0.05% Tween-20 [PBST] and 10% fetal bovine serum), followed by serial two-fold dilutions of each sample in duplicates across the plate. A starting dilution of 1:160, 1:80, and 1:20 was used for serum (IgA and IgG), BALF (IgG) and nasal wash and BALF (IgA) samples, respectively. Plates were incubated with the test serum samples for 2 h at 37 °C. After rigorous plate washes in a microplate washer, plates were incubated with anti-hamster antibodies. For IgG ELISA, a 1:4000 dilution of an HRP-conjugated goat anti-hamster IgG (6060-05, Southern Biotech, Birmingham, Alabama) was added to assay wells. For IgA ELISA, a rabbit anti-hamster IgA antibody [sandwich antibody; (cat. #sab 3001a) Brookwood Biomedical, Jemison, Alabama] was added to assay wells at 1:4000 dilution and plates incubated for 1 hour at 37 °C. Unbound sandwich antibody was washed off and a 1:4000 dilution of an HRP-conjugated goat anti-rabbit IgG (4030-05, Southern Biotech, Birmingham, Alabama) was added to assay plates. In both IgG and IgA ELISAs, incubation with HRP-conjugated secondary antibodies lasted 1 hour after which plates were rigorously washed to remove unbound antibodies. The ABTS/$H_2O_2$ peroxidase substrate (SeraCare, Gaithersburg, Maryland) was added to assay wells and plates left at room temperature for 20 to 30 minutes. Color development was stopped by adding 1% SDS and $OD_{405}$ values were captured on the VersaMax microplate reader with Softmax Pro 7 software version 7.0, build 226962 (Molecular Devices). In the IgG ELISA, the mean $OD_{405}$ values of PBS treatment groups were subtracted from the mean $OD_{405}$ values from other treatment groups and the assay endpoint was a mean $OD_{405}$ value 0.05 (i.e., after background subtraction). In the IgA ELISA, the assay endpoint was a mean $OD_{405}$ value 0.02 of duplicate wells. Antibody titer was defined as the reciprocal of the highest

dilution of a sample at which the mean $OD_{405}$ value for duplicate wells was 0.02 (IgA) or 0.05 after background subtraction (IgG).

## IFN-gamma ELISpot

Hamster interferon gamma (IFN-γ) enzyme-linked immunosorbent spot (ELISpot) analysis was performed using the Hamster IFN-γ ELISpotBASIC (MABTECH Mabtech 3102-2H, Nacka Strand, Sweden) kit according to the manufacturer's instructions. MSIP Plates (Millipore) were washed 5 times with sterile water, coated with mAb (MTH21) and incubated overnight at 4 °C. Coated plates were washed 5 times with 1X PBS, blocked for 30 minutes (room temperatures) with supplemented RPMI 1640 (GibcoBRL) containing 10% heat inactivated FBS, 1% 100x penicillin, streptomycin, and L-Glutamine solution (GibcoBRL). In all, $2.5 \times 10^5$ freshly isolated splenocytes were seeded in each well and stimulated for 45-48 hours at 37 °C with SARS CoV-2 Spike proteins peptide pools (2 μg/ml each peptide) (BEI Catalog No. NR-52418) or Nucleocapsid proteins peptide pools (2 μg/ml each peptide) (BEI Catalog No. NR-52419) prepared in serum free RPMI 1640. Negative and positive plate controls were medium or 2 μg/ml concanavalin A (ConA, Sigma-Aldrich), respectively. Plates were incubated with 1 μg/ml mAb (MTH29-biotin) for 2 h, and then 1 h with Streptavidin-HRP, and finally developed after adding TMB substrate (product No. 3651-10). Distinct spots typically emerge within 20 minutes. After drying, spots were counted using a BioTek Cytation 7 imaging reader (Agilent) and analysis software Gen5 Version No. 3.11. ELISpot data was analyzed in Microsoft excel. The average number of spots from two negative wells (unstimulated cells) was subtracted from peptide pools stimulated wells for each plate. Results were expressed as difference in spots forming cells (SFC)/$10^6$ PBMC between negative control and peptide pools stimulations conditions. Results were plotted using GraphPad Prism 9.4.

## Lung immunofluorescence analyses

Formalin-fixed paraffin-embedded (FFPE) lung sections 4 μm thick were dewaxed, rehydrated, and heat-treated in a microwave oven for 15 min in 10 mM Tris/1 mM EDTA buffer (pH 9.0). After cooling for 30 min at room temperature, heat-retrieved sections were blocked in PBST with 2.5% bovine serum albumin (BSA) for 30 min at RT followed by overnight incubation at 4 °C with primary antibodies in 1% BSA. Primary antibodies used included SARS nucleocapsid protein (NP) (1:800, Sino Biologicals, 40143-MM05), MX1 (1:100, Proteintech, 13750-1-AP), prosurfactant protein C (ProSPC) (1:200, EMD Millipore, AB3786), Iba1 (1:100, Abcam, ab5076), RAGE (1:400, Abcam, ab216329), and E-cadherin (ECAD) (1:50, Abcam, ab219332). Sections were rinsed and incubated with Alexa Fluor 488 (1:200, A-21206) and Alexa Fluor 647-conjugated secondary antibodies (1:200, A-31571 and A-21447) for 1 hour at RT (ThermoFisher, Waltham, MA). Nuclei were counterstained with Hoechst 33342. For double labeling experiments, primary antibodies were mixed and incubated overnight at 4 °C. For negative controls, sections were incubated without the primary antibody or mouse and rabbit isotype antibody controls. Sections stained with conjugated secondary antibodies alone showed no specific staining. Whole slide fluorescence imaging was performed using a Hamamatsu NanoZoomer 2.0-RS whole-slide digital scanner equipped with a 20x objective and a fluorescence module #L11600. Analysis software NDP.view2 (version 2.7.52) was used for image processing (Hamamatsu Photonics, Japan). Immunofluorescence and differential interference images were also captured using an Axio Observer Z1 inverted microscope (Carl Zeiss, Thornwood, NY) equipped with an Axiocam 506 monochrome camera, an ApoTome.2 optical sectioning system, and a Plan-Apochromat 63x/1.4NA oil immersion with WD = 0.19 and Plan-Apochromat 20x/0.8 objective lens. Digital image post-processing and analysis were performed using the ZEN 2 ver. 2.0 imaging software. Images were constructed from Z-stack slices collected at 0.48 μm intervals (4 μm thickness in total) and visualized as maximum intensity projections in orthogonal mode. For semi-quantitative analysis of NP staining, high resolution whole-slide digital images of each lung section were acquired and the NDP.view2 software was used to measure the NP-stained area as a percentage of total area of the section. For TUNEL staining, sections were deparaffinized, hydrated, and pretreated with Proteinase K, followed by EDTA, distilled $H_2O$ wash, and BSA blocking. Sections were then incubated in a reaction mixture (TdT, dUTP, and buffer), washed, and incubated with anti-digoxigenin antibody (1:100, #11093274910, Roche Molecular Biochemicals, Indianapolis, IN). Sections were then visualized with alkaline phosphatase-ImmPACT Vector Red and counterstained with hematoxylin.

## SARS-CoV-2 pseudovirus production and infection assay

Human codon-optimized cDNA encoding SARS-CoV-2 S glycoprotein of the WA1/2020 and the BA.5 variant (with c-terminal 19 amino acids deleted) were synthesized by GenScript and cloned into eukaryotic cell expression vector pcDNA 3.1 between the BamHI and XhoI sites. Pseudovirions were produced by co-transfection of Lenti-X 293T cells with psPAX2, pTRIP-GFP, and SARS-CoV-2 S expressing plasmid using Lipofectamine 3000. The supernatants were harvested at 48 and 72 hours post-transfection and filtered through 0.45-μm membranes. Infection was done as previously described[26]. Forty-eight hours post-infection, cells were fixed and imaged by Leica Stellaris 5 confocal microscope. Nuclei were stained with 4′,6-diamidino-2-phenylindole (DAPI).

## Boosting of MVA-vaccinated BALB/c mice with LAV

Adult male BALB/c mice were purchased from Jackson Laboratory (Bar Harbor, ME). The animals were housed in groups of up to 5 in sealed, individually ventilated cages. Cages were maintained under controlled light (12:12 Light/Dark cycle), temperature (72 °F ± 1 °F), and humidity (30–70%) conditions. Mice were fed and watered *ad libitum* with environmental enrichment, and all animals were acclimatized for 4–6 days or more prior to the experiments. BALB/c mice first received two doses of vaccinia virus Ankara vectors expressing full-length WA1 Spike, ie.,MVA-S[27] in the ABSL-2 facility and then transferred to the ABSL-3 facility where $10^4$ PFU BA.1-LAV or BA.5-LAV were intranasally administered to these mice 8 weeks after the second dose of MVA-S. The study protocol details were approved by the White Oak Consolidated Animal Care and Use Committee and carried out in accordance with the PHS Policy on Humane Care & Use of Laboratory Animals (ASP 2020-30, ASP 2008-02).

## Statistical analysis

One-way ANOVA or Student *t* test was used to calculate statistical significance through GraphPad Prism (9.4.0) software for Windows, GraphPad Software, San Diego, California USA, www.graphpad.com.

## Reporting summary

Further information on research design is available in the Nature Portfolio Reporting Summary linked to this article.

# Data availability

No microarray, DNA sequencing, RNA-seq or proteomic datasets were included in this study. No clinical datasets or third-party data were included. All unique/stable reagents generated in this study are available from the corresponding author with a completed Materials Transfer Agreement. Source data are provided with this paper. Datasets generated and/or analyzed during the current study are appended as supplementary data. Source data are provided with this paper.

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

## Acknowledgements

We would like to thank the White Oak FDA Animal Program Staff and Veterinarians for assistance in conducting this work. The following reagents were obtained through Biodefense and Emerging Infections Research Resources Repository, NIAID, NIH: SARS-Related Coronavirus 2, Isolate USA-WA1/2020, NR-52281; Isolate hCoV-19/USA/MD-HP05647/2021 (Lineage B.1.617.2; Delta variant), NR-55672, contributed by Dr. Andrew S. Pekosz; Isolate hCoV-19/USA/HI-CDC-4359259-001/2021 (Lineage B.1.1.529; Omicron Variant, BA.1), NR-56475, contributed by Centers for Disease Control; Isolate hCoV-19/USA/NY-MSHSPSP-PV56475/2022 (Lineage BA.2.12.1; Omicron Variant), NR-56781, deposited by Dr. Viviana Simon; Isolate hCoV-19/USA/COR-22-063113/2022 (Lineage BA.5; Omicron Variant), NR-58616, contributed by Dr. Richard J. Webby. We thank Drs. A. Rosenfeld and A. Kachko for critical reading the manuscript. The work described in this manuscript was supported by U.S. Food and Drug Administration (FDA) intramural research fund. K.S. was supported by the U.S. FDA, Office of the Chief Scientist, Medical Countermeasures Initiative, under OCET Intramural Program ID# 2022-1821. The funders had no role in study design, data collection and analysis, decision to publish, or preparation of the manuscript. The content of this publication does not necessarily reflect the views or policies of the Department of Health and Human Services, nor does mention of trade names, commercial products, or organizations imply endorsement by the US Government.

## Author contributions

Conceptualization: T.T.W. Methodology: C.B.S., F.D., P.S., C.A.M., S.L., and T.T.W. Investigation: C.B.S., P.S., F.D., C.A.M., C.L.P., S.L., K.S., C.Z.L., M.F.S., J.P.W., and T.T.W. Visualization: C.B.S., F.D., and T.T.W. Funding acquisition: F.D. and T.T.W. Project administration: T.T.W. Supervision: J.P.W. and T.T.W. Writing: C.B.S., F.D., and T.T.W.

## Competing interests

C.B.S., P.S., S.L., C.Z.L., and T.T.W. are inventors on a patent being filed by the U.S. FDA based on the results described in this manuscript. The remaining authors declare no competing interests.
