## [Peer Review File · Nature Communications]

Intranasal or airborne transmission-mediated delivery of an attenuated SARS-CoV-2 protects Syrian hamsters against new variantsREVIEWER COMMENTS

Reviewer #1 (Remarks to the Author):

The team around Dr. Stauff and colleagues presents data on a live-attenuated vaccine (LAV) candidate for COVID. Their LAV candidate is based on an ancestral 2019/2020 SARS-CoV-2 clone. The present study is a follow-up study on another recently published data set on the same LAV.

Immunogenicity and efficacy data were generated in the hamsters model of SARS-CoV-2 infection, using early prototype pathogenic virus strain WA1/2020 as comparator (immunization by pre-exposure) and VoC Delta and Omicron BA1 and BA2 as challenge viruses. As expected, the level of protection achieved against these VoC is dramatically reduced.

I judge the elegant methodology used as a strong asset of the study, making an effort to use and integrate the expanding toolbox in this emerging small animal model, with as unique points (i) measurement of secretory IgA as proxy for mucosa-associated immunity and (ii) CMI by ELISpot including for the viral N protein.

Main concerns:

(1) In a somewhat rough judgement, the authors show that their LAV that is based on prototypic 2019/2020 virus is failing equally bad regarding protecting from emerging VoC as pre-exposure by earlier virus strains, and basically like any first-generation vaccine based on prototypic S sequences. Why were these data not shown with the previous MS to demonstrate the need to update antigens, even such strong platform as the LAV presented by the team?

I highly appreciate the conclusion of a section on "limitations of the study". However, I believe it is not appropriate to not present a way forward for a variant-proof antigen. The authors may need to seriously discuss how they want to update their COVID-19 vaccine candidate for future variants, ideally with showing best evidence for improvement.

(2) Unfortunately, the correlation between the different parameters measured (nAb, IgG, SIgA, SFU, etc.) with protection from pathology are not clearly defined and described (Fig. S2). Can the authors based on their findings define an true added value of SIgA? All correlation are based on low numbers and in NW animals, which all were not protected, SIgA titers appear suspiciously high as early as 2 DPI. I suggest a more vigorous analysis and discussion.

(3) Future COVID vaccines will be booster vaccines (I thus honestly disagree with the concluding statement on the use of the presented LAV as primary vaccine)? What evidence do the authors have that current mRNA vaccines, or previous virus exposure does not ablate immunogenicity of the LAV? I would love to see that hamsters exposed to WA1/2020 can still be boosted with the LAV. In my understanding, pre-existing mucosal immunity is the main reason for a drop in vaccine efficacy in adults (compared to children) for LAV flu vaccine Flumist. Does similar obstacles await your LAV COVID vaccine candidate as well?

Do pre-exposed hamsters boosted with the LAV still shed? This may be indirect evidence for the role of SIgA in protection from infection and transmission/shedding.

(4) Data from longevity of protection are missing, also because mucosal immunity may wain relatively fast, in particular following such a low dose exposure (100 PFU) and using a LAV rather than a pathogen.

(5) Regarding terminology, "passive immunization" is not the correct term for what happend in the experiments showing aerosol transmission of the LAV. "Passive immunization" is generally defined as immunity conferred by administration of e.g. antibodies/serum/monoclonal/T-cells derived from an actively immunised (vaccinated) donor to another individual. In fact vaccinated animals shed virus that can lead to airborne infection of naive animals. Thus I believe, this aspect of deliberate release of a modified live virus is to be considered more as safety-related study. Notably, uncontrolled vaccination (population-wide shedding) may harm people for whom LAV vaccination may be contra-indicated, such as Beverly immune-compromised.

(6) Please mention more clearly in the titel, abstract, main text and figure legends that the LAV under study is a "candidate".

Minor comments:

Line 71: REF19 is a commercial press release. I think it is not appropriate to support main hypothesis by referring to solely commercial, non-peer-reviewed material, I.e. why nasal vaccination may be better than intramuscular.

Line 78: ... efficacy in protecting <from> disease (singular), ...

Line 95: GMT are given throughout. Can IQR or similar measures of variation be added.

Line 429-430: Construction and characterisation of of H1299-hACE2 is not properly described

Line 441-446: Virus stocks were deep sequenced for identity. A statement is missing regarding presence/absence/percentage of S1/S2 cleavage site and other tissue culture adaptive mutations.

Line 449: Please specify age of animals.

Line 460: In previous paragraph both sexes were mentioned. What is the difference here. To which experiments does this paragraph relate?

Line 483: Please specify peptide pools, e.g. listing catalog numbers. The specifications mentioned here are different from those mentioned in Lines 577-78. Please clarify.

Line 490: How was LoD defined/calculated.

Line 591: 4µm sections? In Line 497 5µm are mentioned, please resolve inconsistency. Item in line 613-614, 5µm.

In Fig-7c,d LAV peaks later in both donors and sentinels. Is the virus really attenuated, or does it cause simply a protracted course of infection which would be of major concern? It would be intriguing to see viral loads for later time points beyond 5 DPI.

In most Figures significance levels are only specified in comparison to non-vaccinated. Can the values LAV and WA also be compared?

Reviewer #2 (Remarks to the Author):

Summary:

This study by Stauft et al. is further evaluation of an intranasal live attenuated SARS-CoV-2 vaccine in the Syrian Hamster model. Specifically, this study compares levels of systemic and mucosal immunity in response to the attenuated virus as compared to the ancestral WA1 strain and examines levels of protection against emerging variants of concern (Delta and Omicron). Further, the authors examined the ability of passive infection/immunization of the attenuated virus in eliciting a protective immune response against relevant VOCs in recipient sentinel hamsters. This data represents an important addition to the previously published work in citation 23.

Major critiques:

1. As admitted by the authors in the limitations section, Omicron BA1 and BA2 strains pathology appears to be quite low in Syrian hamsters, making it difficult to fully interpret protective efficacy against the newest VOCs in this particular animal model.

Minor critiques:

1. Line 131: "great than" should be "greater than"
2. Line 146-147: Confusing wording that makes it difficult to understand the group definitions and sizes. I believe you want to move "along with unvaccinated controls (n=8 per group) to the end of the sentence. Or some sort of rewording that makes things more clear.
3. Line 146 and line 844 (Figure 3) seem to have different group size. Is it n=6 or n=7?
4. Line 249: should be "days 3-9"
5. Similarly, line 268 "days 4-5".

Summary of main changes in the revision

1. We have supplied a new Fig. 9 summarizing results from a challenge study in which we found that our prototype Nsp1-K164A/H165A-vaccinated animals were protected against BA.5 challenge in both upper and lower respiratory tract. Notably, this BA.5 isolate caused more severe disease in Syrian hamsters compared to the BA.1 and BA.2.12.1 isolates.
2. We have supplied a new Fig. 10 to demonstrate that attenuated LAV bearing BA.1 and BA.5 spike readily boosted the neutralizing antibodies against BA.1 and BA.5, respectively, in mice that were previously vaccinated with a MVA vector expressing the ancestral spike protein.
3. We have also revised relevant figures and text to improve clarity and statistics.

A point-by-point response to the referee comments is provided as below (original review comments in black and our responses in blue). The line numbers cited in this response correspond to those in the revised manuscript with track changes.

Reviewer #1 (Remarks to the Author):

The team around Dr. Stauff and colleagues presents data on a live-attenuated vaccine (LAV) candidate for COVID. Their LAV candidate is based on an ancestral 2019/2020 SARS-CoV-2 clone. The present study is a follow-up study on another recently published data set on the same LAV. Immunogenicity and efficacy data were generated in the hamsters model of SARS-CoV-2 infection, using early prototype pathogenic virus strain WA1/2020 as comparator (immunization by pre-exposure) and VoC Delta and Omicron BA1 and BA2 as challenge viruses. As expected, the level of protection achieved against these VoC is dramatically reduced.

I judge the elegant methodology used as a strong asset of the study, making an effort to use and integrate the expanding toolbox in this emerging small animal model, with as unique points (i) measurement of secretory IgA as proxy for mucosa-associated immunity and (ii) CMI by ELISpot including for the viral N protein.

Response: We thank the reviewer for these positive comments regarding the strength and the novelty of our work.

Major comments:

In a somewhat rough judgement, the authors show that their LAV that is based on prototypic 2019/2020 virus is failing equally bad regarding protecting from emerging VoC as pre-exposure by earlier virus strains, and basically like any first-generation vaccine based on prototypic S sequences. Why were these data not shown with the previous MS to demonstrate the need to update antigens, even such strong platform as the LAV presented by the team?

I highly appreciate the conclusion of a section on “limitations of the study”. However, I believe it is not appropriate to not present a way forward for a variant-proof antigen. The authors may

need to seriously discuss how they want to update their COVID-19 vaccine candidate for future variants, ideally with showing best evidence for improvement.

Response: We thank the reviewer for these helpful comments. We would like to point out that our prototypic LAV-vaccinated Syrian hamsters remain protected against all the new variants that were tested in this study, including the more pathogenic BA.5 variant (added to this revision as new Figure 9). Collectively, these findings support that this LAV vaccine provides a degree of protection that differs than that of first-generation vaccines based on prototypic S sequences.

The efficacy of our LAV candidate is on a par with that of natural immunity acquired from previous infection, at least in Syrian hamsters (Liu et al. Nat Commun, 2022; Stauff et al. JMV, 2022; Halfmann et al., 2022, Cell Reports). A recent study has shown that immunity acquired from previous infection was more protective than mRNA vaccination against re-infection with the BA.1 variant in hamsters (Halfmann et al., 2022, Cell Reports). Notably, protection was observed with only marginal levels of anti-BA.5 neutralizing antibodies in hamsters vaccinated with our prototype LAV bearing the WA1 spike protein. Nonetheless, these animals were protected against BA.5 at both upper and lower respiratory tract. In this regard, at least in the Syrian hamster model, it is difficult to evaluate the efficacy of the so-called “variant proof vaccine”, partially because convalescent hamsters from a previous infection develop nearly life-long protective immunity against heterologous SARS-CoV-2 variants as observed by us and many others. This also highlights the significant advantage of LAV over spike-based subunit vaccines in that LAV elicits broader immunity that targets multiple viral components, resulting in better protection.

To further demonstrate the expandability of our LAV platform, we conducted new experiments in mice to test if a LAV bearing a different spike may be administered as a booster over animals that are previously vaccinated using a S subunit vaccine. As shown in new Fig. 10, attenuated SARS-CoV-2 bearing BA.1 and BA.5 spike readily boosted neutralizing antibodies against BA.1 and BA.5, respectively, in mice that were previously vaccinated with a MVA vector expressing the ancestral spike protein.

The following text has been added to the revised manuscript:

Line 32 (Abstract): “Similarly attenuated viruses bearing BA.1 and BA.5 spike boosted variant-specific neutralizing antibody titers in mice that were first vaccinated with vaccinia virus Ankara vectors expressing full-length WA1 Spike.”

Line 82 (Introduction): “...possible development into a variant-specific booster...”

Line 336 (Results): “**Variant-specific Nsp1-K164A/H165A as a booster.** To test if Nsp1-K164A/H165A may be used as a booster vaccine candidate, we generated two additional attenuated viruses with the WA1 spike being replaced with either BA.1 or BA.5 spike protein, namely BA.1-LAV and BA.5-LAV, respectively (Fig. 10a). Notably, we have previously published that BA.1 enters cells expressing mouse ACE2 (mACE2) and infects laboratory mouse strain such as Balb/c²⁹. In this study, we also confirmed that

BA.5 infects 293T cells expressing mACE2 (Fig. 10b). Hence, we envisioned that BA.1-LAV and BA.5-LAV would infect Balb/c and induce variant-specific antibody response. To test this possibility, Balb/c mice first received two doses of vaccinia virus Ankara vectors expressing full-length WA1 Spike (MVA-S)³⁰. After 8 weeks, 104 PFU BA.1-LAV or BA.5-LAV were intranasally administered to these mice (Fig. 10c)²⁹. Nonetheless, a dose of 104 PFU was chosen to ensure that the candidate vaccine virus infects the mice with pre-existing immunity. Comparison between pre- and post-boost sera (2 weeks) revealed that pre-boost sera contained high levels of nAbs against the original WA1 isolate, but nondetectable anti-BA.1 or anti-BA.5 nAb. Importantly, two weeks after one dose of BA.1-LAV and BA.5-LAV, the GMT values of corresponding nAB titers increased to 143 (IQR 387.6) and 84 (IQR 315.9), respectively (Fig. 10d)."

Line 404 (Discussion): "Most intriguingly, attenuated SARS-CoV-2 bearing BA.1 and BA.5 spike readily boosted the neutralizing antibody titers against BA.1 and BA.5, respectively, in mice that were previously vaccinated with a MVA vector expressing the spike protein of the ancestral Wuhan isolate."

Line 1142 (Legends): "**Fig. 10. BA.1-LAV and BA.5-LAV as boosters.** **a** Genome organization of BA.1-LAV and BA.5-LAV. Leader sequence (red), transcriptional regulatory sequence within the leader sequence and within the body are highlighted as green bars. The polybasic insert "HRRR" was removed from the spike protein proteins and ORF6-8 were removed from the WA1/2020 backbone. Locations of K164A/H165A are highlighted in the figure. **b** Pseudovirus bearing BA.5 spike infected 293-mACE2 cell line. Infected cells are shown in green. **c** Overall study design to test BA.1-LAV and BA.5-LAV as boosters. 10-week-old male Balb/c mice were employed in this study. **d** Neutralizing antibody titers were measured from pre- and post-boost mouse sera. Each solid circle represents one animal. Numbers above each group indicate the geometric means of neutralizing antibody titers."

Unfortunately, the correlation between the different parameters measured (nAb, IgG, SIgA, SFU, etc.) with protection from pathology are not clearly defined and described (Fig. S2). Can the authors based on their findings define an true added value of SIgA? All correlation are based on low numbers and in NW animals, which all were not protected, SIgA titers appear suspiciously high as early as 2 DPI. I suggest a more vigorous analysis and discussion.

Response: After carefully considering the Reviewer's comments, we have decided to remove Fig. S2 from the manuscript due to insufficient power for meaningful statistical analysis.

Future COVID vaccines will be booster vaccines (I thus honestly disagree with the concluding statement on the use of the presented LAV as primary vaccine)? What evidence do the authors have that current mRNA vaccines, or previous virus exposure does not ablate immunogenicity of the LAV? I would love to see that hamsters exposed to WA1/2020 can still be boosted with the LAV. In my understanding, pre-existing mucosal immunity is the main reason for a drop in vaccine efficacy in adults (compared to children) for LAV flu vaccine Flumist. Does similar obstacles await your LAV COVID vaccine candidate as well?

Response: As shown in new Fig. 10, attenuated SARS-CoV-2 bearing BA.1 and BA.5 spike readily boosted serum neutralizing antibodies against BA.1 and BA.5, respectively, in mice that were previously vaccinated with a MVA vector expressing the spike protein of the Wuhan-variant.

Do pre-exposed hamsters boosted with the LAV still shed? This may be indirect evidence for the role of SIgA in protection from infection and transmission/shedding.

Response: We have previously observed and emphasized that sterile immunity in the nasal cavity is probably an unrealistic goal for a COVID vaccine (Liu et al. Nat Commun, 2022). Our position on this subject has not changed, i.e., reduction in transmission/shedding, but not transmission blocking, is a more appropriate goal for a vaccine, especially when the time window of effectiveness is considered. Our LAV-vaccinated hamsters shed very little virus upon challenge (at least 2-3 logs reduction compared to unvaccinated animals, see Fig. 9), which makes it even harder to quantify how much less virus shedding a booster will do.

Data from longevity of protection are missing, also because mucosal immunity may wain relatively fast, in particular following such a low dose exposure (100 PFU) and using a LAV rather than a pathogen.

Response: We and others have demonstrated that convalescent hamsters display high levels of serum neutralizing antibodies after 7 months and 21 months, respectively (Stauft et al. JMV, 2022 and Halfmann et al., 2022, Cell Reports). However, we agree with the reviewer that mucosal immunity may wane relatively faster, and this is a topic that we are currently pursuing. The following text has been added to the revised manuscript: Line 388 The durability of the mucosal IgA response is currently under investigation.

Regarding terminology, “passive immunization” is not the correct term for what happend in the experiments showing aerosol transmission of the LAV. “Passive immunization” is generally defined as immunity conferred by administration of e.g. antibodies/serum/monoclonal/T-cells derived from an actively immunised (vaccinated) donor to another individual. In fact vaccinated animals shed virus that can lead to airborne infection of naive animals. Thus I believe, this aspect of deliberate release of a modified live virus is to be considered more as safety-related study. Notably, uncontrolled vaccination (population-wide shedding) may harm people for whom LAV vaccination may be contra-indicated, such as Beverly immune-compromised.

Response: The reviewer raises a valid point. To avoid confusion, we have amended the title to read “Active and passive vaccination of Syrian hamsters with an attenuated SARS-CoV-2 protects against new variants of concern”.

Please mention more clearly in the titel, abstract, main text and figure legends that the LAV under study is a “candidate”.

Response: Relevant text passages have been amended as suggested by the reviewer.

Line 83 (Introduction): "...SARS-CoV-2 vaccine candidate."

Line 337 (Results): "...used as a booster vaccine candidate"

Line 379 (Discussion) "...immunogenicity of SARS-CoV-2 vaccine candidates..."

Line 397 (Discussion): "The efficacy of our LAV candidate..."

Line 402 (Discussion): "In case the composition of the LAV candidate..."

Specific minor comments

Line 71: REF19 is a commercial press release. I think it is not appropriate to support main hypothesis by referring to solely commercial, non-peer-reviewed material, I.e. why nasal vaccination may be better than intramuscular.

Response: Reference #19 has been replaced with the following peer-reviewed publication.

19. Wang, Y. et al. Scalable live-attenuated SARS-CoV-2 vaccine candidate demonstrates preclinical safety and efficacy. Proc Natl Acad Sci U S A 118 (2021). <https://doi.org/10.1073/pnas.2102775118>

Line 78: ... efficacy in protecting <from> disease (singular), ...

Response: This sentence has been amended (Line 81).

Line 95: GMT are given throughout. Can IQR or similar measures of variation be added.

Response: The Results section of the revised manuscript has been amended to include interquartile ranges where appropriate.

Line 429-430: Construction and characterisation of H1299-hACE2 is not properly described

Response: As described in the Materials and Methods (Line 468), H1299-hACE2 is a human lung carcinoma cell line stably expressing human ACE2. The cell line was generated by lentiviral transduction of the NCI-1299 human lung carcinoma cell line (ATCC CRL-5803) with pLVX-hACE2 and selected with 1 µg/mL puromycin. A western blot was performed to confirm the expression of hACE2. H1299-hACE2 cells were maintained in DMEM supplemented with 5% penicillin and streptomycin, and 10% fetal bovine serum (FBS) at 37 °C with 5% CO₂.

Line 441-446: Virus stocks were deep sequenced for identity. A statement is missing regarding presence/absence/percentage of S1/S2 cleavage site and other tissue culture adaptive mutations.

Responses: A statement has been added to address this comment. Line 482 now reads "Passaged viruses were deep sequenced to confirm identity (100% match with the original

sequence, i.e., free of tissue culture adaptive mutations such as the loss of the polybasic site between S1 and S2 subunit of the spike protein).”

Line 449: Please specify age of animals.

Responses: Ages of animals are provided in the relevant text of Materials and Methods (Line 496) or the Figure 10 legend (line 1150).

Line 460: In previous paragraph both sexes were mentioned. What is the difference here. To which experiments does this paragraph relate?

Responses: Only male mice were used in this study. We deleted “and female” (Line 496).

Line 483: Please specify peptide pools, e.g. listing catalog numbers. The specifications mentioned here are different from those mentioned in Lines 577-78. Please clarify.

Responses: We thank the reviewer for pointing out this error. The information has been added as “BEI Catalog No. NR-52418 and NR-52419” (Line 531).

Line 490: How was LoD defined/calculated.

Responses: Limits of detection for the FFA were set based on the minimum detectable titer given 1 focus-forming unit at the lowest dilution (10^{-1}) and inoculation volume (50 μ l). Any values below the lower limit (200 FFU/ml) were arbitrarily assessed as 100 FFU/ml for statistical analysis. The method for assigning limits of quantification for qRT-PCR are described in the references listed in the Materials and Methods section and are based on the linearity of plasmid-based standards for the target sequence.

35. Selvaraj, P. et al. SARS-CoV-2 infection induces protective immunity and limits transmission in Syrian hamsters. *Life Sci Alliance* 4 (2021).

44. Stauff, C. B., Lien, C. Z., Selvaraj, P., Liu, S. & Wang, T. T. The G614 pandemic SARS-CoV-2 variant is not more pathogenic than the original D614 form in adult Syrian hamsters. *Virology* 556, 96-100 (2021).

Line 591: 4 μ m sections? In Line 497 5 μ m are mentioned, please resolve inconsistency. Item in line 613-614, 5 μ m.

Responses: Tissues used for H&E stain were sectioned at 5 μ m per slice. Those for immunofluorescence analyses were sectioned at 4 μ m per slice.

In Fig-7c,d LAV peaks later in both donors and sentinels. Is the virus really attenuated, or does it causes imply a protracted course of infection which would be of major concern? It would be intriguing to see viral loads for later time points beyond 5 DPI.

Responses: The attenuation of the vaccine virus has been extensively characterized both *in vitro* and *in vivo* (Liu et al. Nat Commun, 2022). Infectious virus titers in nasal wash samples generally subside after 4 days post-intranasal inoculation (Fig. 7c). DPE in Fig. 7d represents days post-exposure. In the airborne transmission model, the growth kinetics of SARS-CoV-2 in nasal cavity is delayed in comparison to direct intranasal inoculation (Port et al. Nat Commun. 2021). Nonetheless, it does not represent a protracted infection. It just takes longer for the virus to establish the infection in such a model. With an attenuated virus, it took a couple of days for the transmission to occur.

In most Figures significance levels are only specified in comparison to non-vaccinated. Can the values LAV and WA also be compared?

Responses: In most studies where both WA1 and LAV are present, statistical analyses revealed no significant differences between these two groups. Hence, this was not shown on the relevant figures to simplify their appearance (Figs. 3,4,5).

Reviewer #2 (Remarks to the Author):

This study by Stauft et al. is further evaluation of an intranasal live attenuated SARS-CoV-2 vaccine in the Syrian Hamster model. Specifically, this study compares levels of systemic and mucosal immunity in response to the attenuated virus as compared to the ancestral WA1 strain and examines levels of protection against emerging variants of concern (Delta and Omicron). Further, the authors examined the ability of passive infection/immunization of the attenuated virus in eliciting a protective immune response against relevant VOCs in recipient sentinel hamsters. This data represents an important addition to the previously published work in citation 23.

Response: We thank the reviewer for the positive feedback.

Other major concerns:

1. As admitted by the authors in the limitations section, Omicron BA1 and BA2 strains pathology appears to be quite low in Syrian hamsters, making it difficult to fully interpret protective efficacy against the newest VOCs in this particular animal model.

Response: We agree with the reviewer and hence performed an additional challenge study using a more pathogenic BA.5 variant. Shown in the new Fig. 9, BA.5 infection of unvaccinated Syrian hamsters induced weight loss and significant lung pathologies, but the vaccinated animals were protected from developing any of these observable diseases.

We added the following text to the revised manuscript:

Line 318 (Results): **“Nsp1-K164A/H165A vaccination protects against BA.5 challenge.** Because Omicron BA.1 and BA.2.12.1 only caused mild diseases in Syrian hamsters, we performed another vaccination-challenge study using a BA.5 isolate that induces more severe disease. Shown in Fig. 9a, sixty days after vaccination with 100 PFU Nsp1-K164A/H165A bearing the WA1-spike protein, Syrian hamsters were challenged with 104 PFU BA.5 isolate. Unvaccinated animals rapidly lost weight over the next 7 days, whereas the body weight of vaccinated animals remained steady throughout the course of the study (Fig. 9b). Vaccinated animals had detectable virus from nasal wash samples collected at 1 DPC with minimal (at or below) infectious virus at subsequent timepoints. By contrast, unvaccinated animals shed at least 2 logs higher infectious virus in nasal wash samples from 1 to 3 DPC (Fig. 9c). Viral loads in nasal turbinates, BALF, lungs, were nearly undetectable in vaccinated animals at 4 and 7 DPI (Fig. 9d-g). Lastly, Nsp1-K164A/H165A vaccinated hamsters also showed very little lung pathology at 4 and 7 DPC (Fig. 9h-k) in comparison to unvaccinated animals. It is worth mentioning that BA.4/BA.5 and some later Omicron subvariants are even more immune evasive than the initial BA.1 and BA.2 variants, but our prototype Nsp1-K164A/H165A bearing the WA1 spike conferred protection against BA.5 both in the upper and lower respiratory tract in Syrian hamsters.”

Line 389 (Discussion): “Interestingly, Nsp1-K164A/H165A-vaccinated animals were protected in the lung against Omicron BA.1, BA.2.12.1 and even BA.5 challenge despite very little detectable nAbs in these animals against these newer variants.”

Line 1115 (Legends): **“Fig. 9. Intranasal immunization of Syrian hamsters with 100 PFU Nsp1-K164A/H165A is protective against Omicron BA.5 challenge.** **a** Syrian hamsters (male, 5 months old) were vaccinated with 100 PFU Nsp1-K164A/H165A 60 days prior to challenge with 104 PFU BA.5 (isolate hCoV-19/USA/COR-22-063113/2022) (n=5) on day 0. **b** Changes in weight were followed in challenged hamsters 0-7 DPC, with ** indicating $p < 0.01$ and **** indicating $p < 0.0001$. **c** From 1-5 DPC, infectious virus from nasal wash samples was quantified by focus-forming assays for vaccinated and unvaccinated hamsters (n=4). **d-f** Infectious virus titers of nasal turbinates (d), bronchoalveolar lavage fluid (BALF, e), and lung homogenates (f) at 4 and 7 DPC were determined by focus-forming assays. **g** Viral sgRNA levels in lung and nasal turbinate samples from 4 DPC (n=4) were measured by qRT-PCR. Sum clinical scores (**h**) and percentage of consolidation (i) were also compared for lungs collected at 4 and 7 DPC. Heat-map presentation of individual pathologies in lungs collected at 4 DPC (**j**) and 7 DPC (**k**). Graphs for (b) and (g) indicate mean values from a single experiment with standard deviations shown as error bars. Dot plots represent samples collected from individual animals in a single experiment, horizontal bars indicate mean values with standard deviations shown as error bars. Statistical differences were calculated using ordinary one-way analysis of variance (ANOVA) in GraphPad Prism 9.4.0 with Tukey’s multiple comparisons tests and p-values are indicated in the graph where appropriate ($p < 0.05$).”

Minor issues:

1. Line 131: “great than” should be “greater than”

Response: Amended (Line 135).

2. Line 146-147: Confusing wording that makes it difficult to understand the group definitions and sizes. I believe you want to move “along with unvaccinated controls (n=8 per group) to the end of the sentence. Or some sort of rewording that makes things more clear.

Response: We revised the text to read as “the vaccinated and convalescent hamsters (n=6-7 per group) along with unvaccinated naïve hamsters (n=8)” (Line 150).

3. Line 146 and line 844 (Figure 3) seem to have different group size. Is it n=6 or n=7?

Response: For Fig. 1f, the WA1 group had 6 subjects, all other groups contained 7. We amended the text to read as “(n=6-7 per group)” for simplicity (Line 150).

4. Line 249: should be “days 3-9”

Response: We thank the reviewer for noticing the missing word. The text has been amended accordingly (Line 254).

5. Similarly, line 268 “days 4-5”.

Response: The relevant text has been amended (Line 273).

REVIEWERS' COMMENTS

Reviewer #1 (Remarks to the Author):

The authors made a great effort to address all major concerns raised with the previous version. Hereby also my excuses for plenty of typos in my previous assessment. Addition of extra data (including addition of two complete new figures) is appreciated. Boosting by BA5-LAV to broaden immunity towards new VoC is impressive.

Few minor points remain that need to be clarified.

(1) Titel: "Active and passive vaccination ..." I am still not sure if by this wording it is clear what is meant with "passive". This term is not commonly used. In fact what is described in the manuscript is airborne transmission of live vaccine virus between cage mates. Can this still be corrected, please. "Direct vaccination and immunization by airborne transmission of live attenuated SARS-CoV-2 " or similar.

(2) Line 449: Booster immunization with BA5-LAV was done in mice and using a 100-fold higher dose than that used in all previous hamsters studies. Why this sudden switch to a mouse model? How is this justified? This needs to be discussed properly. Was the massive increase in the dose used for booster immunization in pre-immune animals required for vaccine efficacy in these animals? How would the triggered (booster) immunity look like at a lower dose? What about safety? If I am not mistaken, the authors showed previously that a higher dose of LAV may be associated with some pathology. This is a possible limitation of the study that may to be discussed.

(3) The authors forgot to describe the new mouse experiments in the M&M section.

A point-by-point response to the referee comments is provided as below (original review comments in black and our responses in blue).

Reviewer #1 (Remarks to the Author):

The authors made a great effort to address all major concerns raised with the previous version. Hereby also my excuses for plenty of typos in my previous assessment.

Addition of extra data (including addition of two complete new figures) is appreciated. Boosting by BA5-LAV to broaden immunity towards new VoC is impressive.

Few minor points remain that need to be clarified.

(1) Titel: "Active and passive vaccination ..." I am still not sure if by this wording it is clear what is meant with "passive". This term is not commonly used. In fact what is described in the manuscript is airborne transmission of live vaccine virus between cage mates. Can this still be corrected, please. "Direct vaccination and immunization by airborne transmission of live attenuated SARS-CoV-2 " or similar.

Response: the title has been amended to read "Intranasal or airborne transmission-mediated delivery of an attenuated SARS-CoV-2 protects Syrian hamsters against new variants".

(2) Line 449: Booster immunization with BA5-LAV was done in mice and using a 100-fold higher dose than that used in all previous hamsters studies. Why this sudden switch to a mouse model? How is this justified? This needs to be discussed properly. Was the massive increase in the dose used for booster immunization in pre-immune animals required for vaccine efficacy in these animals? How would the triggered (booster) immunity look like at a lower dose? What about safety? If I am not mistaken, the authors showed previously that a higher dose of LAV may be associated with some pathology. This is a possible limitation of the study that may be discussed.

Response: the justification of a booster dosage of 10^4 PFU was already included in the previous submission (lines 331-332): A dose of 10^4 PFU was chosen to ensure that the candidate vaccine virus infects the mice with pre-existing immunity. To further elaborate on this point: BALB/c mice are generally resistant to SARS-CoV-2 infection. Even though Omicron BA.1 and BA.5 have acquired mutations in their spike protein that permit infection via mouse ACE2, we suspected that Balb/c mice are still less susceptible to infection than Syrian hamsters and hence performed the study using a higher dose to ensure a successful vaccination. Previously we have demonstrated that our LAV at 10^4 PFU does not induce lung pathology in Syrian hamsters.

(3) The authors forgot to describe the new mouse experiments in the M&M section.

Response: The M&M section has been updated to include the new mouse experiments.